# OPTIMAL TRANSPORT FOR PROBABILISTIC CIRCUITS

## ABSTRACT

We introduce a novel optimal transport framework for probabilistic circuits (PCs). While it has been shown recently that divergences between distributions represented as certain classes of PCs can be computed tractably, to the best of our knowledge, there is no existing approach to compute the Wasserstein distance between probability distributions given by PCs. We consider a Wasserstein-type distance that restricts the coupling measure of the associated optimal transport problem to be a probabilistic circuit. We then develop an algorithm for computing this distance by solving a series of small linear programs and derive the circuit conditions under which this is tractable. Furthermore, we show that we can also retrieve the optimal transport plan between the PCs from the solutions to these linear programming problems. We then consider the empirical Wasserstein distance between a PC and a dataset, and show that we can estimate the PC parameters to minimize this distance through an efficient iterative algorithm.

## 1 INTRODUCTION

The Wasserstein distance is a statistical distance metric corresponding to the objective value taken by the optimal transport problem as proposed by Kantorovich's optimal transport framework that, given two probability measures, finds its optimal value at a coupling measure where the expected distance between the original two measures is minimized (Kantorovich, 1942). Kantorovich's optimal transport problem is a relaxation of Monge's optimal transport problem, which requires that the probability mass at each point in one distribution not be split up in transport to the second distribution (Monge, 1781). Computing such a distance has proven extremely useful, with applications in generative modeling (Arjovsky et al., 2017), data privacy (Li et al., 2007), and distributionally robust optimization (Rahimian & Mehrotra, 2022). However, computing the Wasserstein distance is a highly intractable task for all but the simplest distributions.

Modeling probability distributions in a way that enables tractable computation of probabilistic queries is of great interest to the machine learning community. Probabilistic circuits (PCs) (Choi et al., 2020) provide a unifying framework for representing many classes of tractable probabilistic models as computational graphs; within this framework, tractability of certain queries can be guaranteed through imposing structural properties on the computational graph of the circuit. This includes tractable marginal and conditional inference, as well as pairwise queries that compare two circuits such as Kullback-Leibler Divergence and cross-entropy (Liang & Van den Broeck, 2017; Vergari et al., 2021). However, to the best of our knowledge, there is no existing algorithm that tractably computes the Wasserstein distance between two probabilistic circuits.

While algorithms for computing other statistical distance measures between PCs are well-established, the Wasserstein distance offers a distinct advantage in many applications. Measures such as KL-divergence and cross-entropy are unbounded between distributions with disjoint supports; conversely, the $p$-Wasserstein distance is always bounded for distributions with finite $p$-th moments (Villani, 2008, p. 107). Computing the Wasserstein distance also provides a bound for other statistical distance metrics such as the Prokhorov metric and the total-variation distance—we suggest (Gibbs & Su, 2002) for more details. However, computing the Wasserstein distance is often more intractable than other probabilistic queries such as the KL-divergence or cross-entropy due to the inherent optimization problem required to be solved. In the case of Boolean distributions, the 1-Wasserstein distance is intimately related to the total-variation (TV) distance; recent work has shown that the latter is intractable to compute exactly, but that efficient approximation algorithms exist (Bhattacharyya et al., 2023; Feng et al., 2023).

This paper focuses on computing (or bounding) the Wasserstein distance and optimal transport plan between (i) two probabilistic circuits and (ii) a probabilistic circuit and an empirical distribution. For (i) we propose a Wasserstein-type distance that upper-bounds the true Wasserstein distance and provide an efficient and exact algorithm for computing it between two circuits (Section 3). For (ii) we propose a parameter estimation algorithm for PCs that seeks to minimize the Wasserstein distance between a circuit and an empirical distribution (Section 4). We empirically evaluate our proposed methods on randomly generated PCs as well as on a benchmark dataset (Section 5).

## 2 Preliminaries

We use capital letters $(X)$ to denote random variables and lowercase letters $(x)$ to denote their assignments. Boldface denotes a set of random variables and their assignments respectively (e.g., $\mathbf{X}$ and $\mathbf{x}$).

### 2.1 Wasserstein Distances and Optimal Transport

Let $P$ and $Q$ be two probability measures on metric space $\mathbb{R}^n$. For $p \geq 1$, the $p$-*Wasserstein distance* between $P$ and $Q$ is defined as:

$$\mathsf{W}_p^p(P, Q) = \inf_{\gamma \in \Gamma(P,Q)} \mathbb{E}_{\gamma(\mathbf{x},\mathbf{y})}[\|\mathbf{x} - \mathbf{y}\|_p^p] \tag{1}$$

where $\Gamma(P, Q)$ denotes the set of all *couplings* which are joint distributions whose marginal distributions coincide exactly with $P$ and $Q$. That is, the following holds for all $\gamma \in \Gamma(P, Q)$:

$$P(\mathbf{x}) = \int_{\mathbb{R}^n} \gamma(\mathbf{x}, \mathbf{y})d\mathbf{y}, \quad Q(\mathbf{y}) = \int_{\mathbb{R}^n} \gamma(\mathbf{x}, \mathbf{y})d\mathbf{x} \tag{2}$$

Here, the *Wasserstein objective* of some (not necessarily optimal) coupling refers to the expectation inside the infimum in Equation 1 taken over that coupling, and the *Wasserstein distance* between two distributions refers to the value taken by the Wasserstein objective for the optimal coupling. It can be shown that there is always a coupling that obtains the infimum above (Villani, 2008). Such optimal coupling $\gamma^*$ induces a *transport plan* $\mathbf{x} \mapsto \gamma^*(\mathbf{x}, .)$. If the coupling is deterministic, this is called a *transport map* $\mathbf{x} \mapsto T(x)$ where $T(x)$ is the support of $\gamma^*(\mathbf{x}, .)$.

### 2.2 Probabilistic Circuits

Many tractable probabilistic models—including arithmetic circuits (Darwiche, 2003), sum-product networks (Poon & Domingos, 2011), cutset networks (Rahman et al., 2014), and more—can be understood through a unifying framework of *probabilistic circuits* (Choi et al., 2020).

**Definition 1** (Probabilistic circuit). A probabilistic circuit (PC) $C$ over a set of discrete or continuous random variables $\mathbf{X}$ is a parameterized, rooted directed acyclic graph (DAG) with three types of nodes: sum, product and input nodes. Each sum node $n$ has normalized parameters $\theta_{n,c}$ for each child node $c$, and each input node $n$ is associated with function $f_n$ that encodes a univariate probability distribution over one of the random variables $X_i \in \mathbf{X}$, also called its *scope* $\mathsf{sc}(n)$. The set of child nodes for an internal node (sum or product) $n$ is denoted $\mathsf{ch}(n)$, and the sub-circuit rooted at any node $n$ parameterizes a probability distribution $p_n(x)$ over its scope $\mathsf{sc}(n) = \bigcup_{c \in \mathsf{ch}(n)} \mathsf{sc}(c)$ defined as follows:[1]

$$p_n(\mathbf{x}) = \begin{cases} f_n(\mathbf{x}) & \text{if } n \text{ is an input node,} \\ \prod_{c \in \mathsf{ch}(n)} p_c(\mathbf{x}) & \text{if } n \text{ is a product node,} \\ \sum_{c \in \mathsf{ch}(n)} \theta_{n,c} p_c(\mathbf{x}) & \text{if } n \text{ is a sum node.} \end{cases}$$

Probabilistic circuits admit exact and efficient computation of many probabilistic inference queries, enabled by enforcing certain structural constraints. In particular, throughout this paper we assume two properties, *smoothness* and *decomposability*, which enable tractable computation of marginal

---

[1]Below, we implicitly project $\mathbf{x}$ onto $\mathsf{sc}(n)$ by only considering the dimensions that correspond to random variables in the node's scope

and conditional queries. A PC is *smooth* if every sum node $n \in C$ satisfies the following: $\forall n_i \in \mathsf{ch}(n)$, $\mathsf{sc}(n_i) = \mathsf{sc}(n)$ (i.e., the children of the sum node have the same scope as the parent); it is *decomposable* if every product node $n \in C$ satisfies the following: $\forall n_i, n_j \in \mathsf{ch}(n)$, $\mathsf{sc}(n_i) \bigcap \mathsf{sc}(n_j) = \emptyset$ (i.e., the children of the product node have disjoint scopes).

# 3 OPTIMAL TRANSPORT BETWEEN PROBABILISTIC CIRCUITS

We now consider the problem of computing Wasserstein distances and optimal transport plans between distributions represented by probabilistic circuits. For notational simplicity, suppose $P(\mathbf{X})$ and $Q(\mathbf{Y})$ are two PCs defining probability measures on a metric space, with a bijective mapping between variables in $\mathbf{X}$ and those in $\mathbf{Y}$; w.l.o.g., let $X_i$ and $Y_i$ map to each other. Moreover, we assume that the univariate input distributions in the PCs allow constant-time computation of the Wasserstein distance, following the standard assumption of tractability of input distributions for tractable inference on PCs. In particular, this is the case for $p$-Wasserstein distance between categorical distributions and for the 2-Wasserstein distance between Gaussian distributions.

Unfortunately, even with the above assumptions, computing the Wasserstein distance between probabilistic circuits is computationally hard.

**Theorem 1.** Suppose $P$ and $Q$ are probabilistic circuits over $n$ Boolean variables. Then computing the $\infty$-Wasserstein distance between $P$ and $Q$ is coNP-hard.

In fact, the above is true even when the PCs satisfy stronger structural constraints (determinism and structured decomposability) that enable tractable inference of hard queries such as maximum-a-posteriori (MAP) Choi & Darwiche (2017) and entropy (Vergari et al., 2021) and even closed-form maximum-likelihood parameter estimtion. At a high level, the proof proceeds by reducing from the problem of deciding consistency of two OBDDs (a type of deterministic and structured-decomposable circuits) which is NP-hard (Meinel & Theobald, 1998, Lemma 8.14). In particular, given the two OBDDs, we can construct two deterministic and structured-decomposable PCs in polynomial time such that the input OBDDs are consistent iff $\mathsf{W}_\infty$ between the PCs is not 1. We refer to Appendix C.1 for a detailed proof.

Theorem 1 shows that computing the $\infty$-Wasserstein distance between two PCs is computationally hard. Whether computing $\mathsf{W}_p$ for some other fixed $p$ (such as $p = 1$ or 2) is NP-hard is still an open question—although there only exist efficient algorithms that bound this quantity between GMMs, rather than compute it exactly (Delon & Desolneux, 2020; Chen et al., 2018)— however, we are interested in efficiently computing or upper-bounding $\mathsf{W}_p$ for *arbitrary* $p$, including $\mathsf{W}_\infty$. Thus, to address this computational challenge, we consider a Wasserstein-type distance between PCs by restricting the set of coupling measures to be PCs of a particular structure. Furthermore, we derive the structural conditions on the input PCs required to construct such structure and find the parameters that minimize the Wasserstein objective in time quadratic in the size of the input circuits.

## 3.1 $\mathsf{CW}_p$: A DISTANCE BASED ON COUPLING CIRCUITS

We propose a notion of *coupling circuit* between two *compatible* (see Definition 2 below) PCs, and introduce a Wasserstein-type distance $\mathsf{CW}_p$ which restricts the coupling set in Equation 1 to be circuits of this form. We then exploit the structural properties guaranteed by coupling circuits, namely smoothness and decomposability, to derive efficient algorithms for computing $\mathsf{CW}_p$ and associated transport plan.

**Definition 2** (Circuit compatibility (Vergari et al., 2021)). Two smooth and decomposable PCs $P$ and $Q$ over RVs $\mathbf{X}$ and $\mathbf{Y}$, respectively, are *compatible* if the following two conditions hold: (i) there is a bijective mapping $\leftrightarrow$ between RVs $X_i$ and $Y_i$, and (ii) any pair of product nodes $n \in P$ and $m \in Q$ with the same scope up to the bijective mapping are mutually compatible and decompose the scope the same way—that is, if $n$ and $m$ have scopes $\mathbf{X}$ and $\mathbf{Y}$ and $\mathbf{X} \leftrightarrow \mathbf{Y}$, then $n$ and $m$ have the same number of children, and for each child of $n$ with scope $\mathbf{X}_i$ there is a corresponding child of $m$ with scope $\mathbf{Y}_i$ such that $\mathbf{X}_i \leftrightarrow \mathbf{Y}_i$. Such pair of nodes are called *corresponding* nodes.

Note that circuit compatibility is necessary for existing tractable algorithms for pairwise queries on PCs (e.g., computing divergences); however, a pair of arbitrary non-compatible circuits may be transformed into two structured-decomposable circuits and then made compatible, albeit with

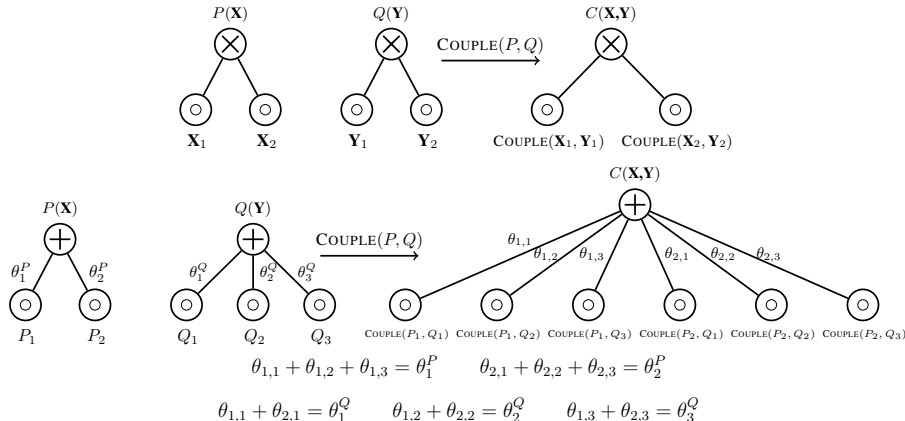

$$\theta_{1,1} + \theta_{1,2} + \theta_{1,3} = \theta_1^P \qquad \theta_{2,1} + \theta_{2,2} + \theta_{2,3} = \theta_2^P$$

$$\theta_{1,1} + \theta_{2,1} = \theta_1^Q \qquad \theta_{1,2} + \theta_{2,2} = \theta_2^Q \qquad \theta_{1,3} + \theta_{2,3} = \theta_3^Q$$

Figure 1: Construction of coupling circuits rooted at corresponding nodes, along with the parameter constraints for a coupled sum node. Product nodes couple children with corresponding scopes, and sum nodes couple the Cartesian product of the children. Note that corresponding sum nodes need not have the same number of children.

a worst-case exponential increase in circuit size (de Colnet & Mengel, 2021; Zhang et al., 2024). Furthermore, compatibility does not require that the circuit structures are the same, only their hierarchical scope partitioning; for example, the number of children of corresponding sum nodes is not constrained (see Appendix A for an example). Lastly, current state-of-the-art PC learning algorithms naturally allow us to learn compatible circuit structures—assuming we assign the bijective mapping ourselves (Dang et al., 2020; Liu & Van den Broeck, 2021).

**Definition 3** (Coupling circuit). A *coupling circuit* $C$ between two compatible PCs $P$ and $Q$ with scopes $\mathbf{X}$ and $\mathbf{Y}$, respectively, is a PC with the following properties. (i) Each node $r \in C$ is recursively a coupling of a pair of nodes $n \in P$ and $m \in Q$.[2] (ii) Each node $r \in C$ that is a coupling of sum nodes $n \in P, m \in Q$ with edge weights $\{\theta_i\}$ and $\{\theta_j\}$ has edge weights $\{\theta_{i,j}\}$ such that $\sum_i \theta_{i,j} = \theta_j$ and $\sum_j \theta_{i,j} = \theta_i$ for all $i$ and $j$.

The second property described above ensures that such coupling circuit $C$ satisfies the marginal-matching constraints in Equation 2 with respect to $P$ and $Q$ (detailed derivation in Appendix C.2). Furthermore, this property ensures that the sub-circuit rooted at any internal node in the coupling circuit matches marginals to the corresponding nodes in the original circuits (which is a stronger constraint than the entire coupling circuit simply matching marginal distributions to the original circuits). We are now ready to define our proposed distance metric between PCs, which is the minimum Wasserstein objective obtained by a valid parameterization of their coupling circuit.

**Definition 4** (Circuit Wasserstein distance). Let $P(\mathbf{X})$ and $Q(\mathbf{Y})$ be compatible PCs and $C_\theta(\mathbf{X}, \mathbf{Y})$ their coupling circuit parameterized by $\theta$. The $p$-*Circuit Wasserstein distance* between $P$ and $Q$ is:

$$\mathsf{CW}_p^p(P, Q) = \min_\theta \mathbb{E}_{C_\theta(\mathbf{x}, \mathbf{y})}[\|\mathbf{x} - \mathbf{y}\|_p^p].$$

We now investigate some properties of $\mathsf{CW}_p$. First, we note that $\mathsf{CW}_p$ is indeed a metric on any set of compatible circuits, which is contrary to some other statistical measures such as KL-divergence used to compare distributions.

**Proposition 1.** For any set $\mathcal{C}$ of compatible circuits, $\mathsf{CW}_p$ defines a metric on $\mathcal{C}$.

Moreover, we have that $\mathsf{CW}$ bounds the true Wasserstein distance between PCs as both are infima of the same Wasserstein objective, while the feasible set of couplings for $\mathsf{CW}$ is more restrictive.

**Proposition 2.** For compatible PCs $P$ and $Q$, $\mathsf{W}_p(P, Q) \leq \mathsf{CW}_p(P, Q)$.

This also implies that the coupling circuit $C(\mathbf{x}, \mathbf{y})$ corresponding to $\mathsf{CW}_p(P, Q)$ induces a (albeit not necessarily optimal) transport plan that maps a point $\mathbf{x}$ to a distribution $C(\mathbf{y}|\mathbf{x})$ and vice versa.

---

[2]The coupling circuit has the same structure as the product circuit (Vergari et al., 2021) of $P$ and $Q$. Informally, this is done by constructing a *cross product* of children at every pair of sum nodes, and the product of corresponding children at every pair of product nodes (see Figure 1). Algorithm 1 shows this construction.

---

**Algorithm 1** COUPLE$(n, m)$: coupling circuit that optimizes $\mathsf{CW}_p^p(n, m)$ of compatible PCs rooted at nodes $n, m$

---

**Note:** We omit calls to a cache storing previously-computed coupling circuits COUPLE$(n, m)$ for simplicity.

1: **if** $n, m$ are input nodes **then** $r \leftarrow$ new product$(n, m)$      $\triangleright$ Product node with children $n,m$
2: **else if** $n, m$ are sum nodes **then**
3:      $r \leftarrow$ new sum node with parameters $\theta_{i,j}$
4:      **for each** $c_i \in n$.children, $c_j \in m$.children **do**
5:         $r$.children$[i, j] \leftarrow$ COUPLE$(c_i, c_j)$
6:      $\mathsf{LP} \leftarrow \begin{cases} \text{minimize} & \sum_i \sum_j \mathsf{CW}_p(r.\text{children}[i,j]) * \theta_{i,j} \\ \text{subject to} & \forall i, \sum_j \theta_{i,j} = \theta_i \\ & \forall j, \sum_i \theta_{i,j} = \theta_j \\ & \theta_{i,j} \in [0,1] \end{cases}$
7:      solve $\mathsf{LP}$                               $\triangleright$ Solve for optimal parameters $\theta_{i,j}$
8: **else if** $n, m$ are product nodes **then**
9:      $r \leftarrow$ new product node
10:      **for each** $c_1 \in n$.children, $c_2 \in m$.children **where** $\mathsf{sc}(c_1) = \mathsf{sc}(c_2)$ **do**
11:         add COUPLE$(c_1, c_2)$ to $r$.children     $\triangleright$ Child is the coupling of corresponding children
12: **return** $r$

---

### 3.2 EXACT AND EFFICIENT COMPUTATION OF $\mathsf{CW}_p$

We now present our algorithm that exactly and efficiently computes the Circuit Wasserstein distance of two compatible PCs, which in turn upper-bounds their Wasserstein distance. The key observation enabling our algorithm is that the Wasserstein objective for a given parameterization of the coupling circuit can be computed recursively through a single feedforward pass through the circuit, and thus can also be minimized over its parameters in a single forward pass.

**Recursive Computation of the Wasserstein Objective** Let $C(\mathbf{X}, \mathbf{Y})$ be a coupling circuit and $g(n) = \mathbb{E}_{C_n}[\|\mathbf{x} - \mathbf{y}\|_p^p]$ the corresponding $\mathsf{CW}_p$-objective function at each node $n \in C$. We can write $g(n)$ recursively as follows (see Appx. C.4 for correctness proof):

$$g(n) = \begin{cases} \mathsf{W}_p^p(c_1, c_2) & \text{if } n \text{ is a product with input node children,} \\ \sum_{c \in \mathsf{ch}(n)} g(c) & \text{if } n \text{ is a product with sum or product node children,} \\ \sum_{c \in \mathsf{ch}(n)} \theta_{n,c} g(c) & \text{if } n \text{ is a sum node.} \end{cases} \quad (3)$$

Thus, we can push computation of the Wasserstein objective down to the leaf nodes of a coupling circuit, and our algorithm only requires a closed-form solution for $\mathsf{W}_p$ between univariate input distributions as the base case. Note that the objective function at a decomposable product node is the *sum* of the objective functions at its children; this is because the $L_p^p$-norm decomposes into the sum of norm in each dimension.

**Recursive Computation of the Optimal Coupling Circuit Parameters for $\mathsf{CW}_p$** Leveraging the recursive properties of the Wasserstein objective, we can compute the optimal parameters of the coupling circuit by solving a small linear program at each sum node. Algorithm 1 details the construction of a coupling circuit and finding the optimal parameters to compute $\mathsf{CW}_p$.

Specifically, we wish to find $\min_\theta g(n)$ where $g(n)$ can be written recursively as in Equation 3. By this definition, at sum nodes we can minimize the Wasserstein objective at each child independently then solve a linear program using the objective value at the child nodes as constants: given the optimal $g(c)$ for each child node $c$ of $n$, we can rewrite $\min_\theta g(n) = \min_\theta \sum_{c \in \mathsf{ch}(n)} \theta_c g(c)$ to see that solving for the sum node parameters reduces to solving a linear program. We can decompose the optimization problem this way because the optimization at children are independent

of the parent parameters. At a product node, we can again push the problem down to the children: $\min_\theta \sum_{c \in \mathsf{ch}(n)} g(c) = \sum_{c \in \mathsf{ch}(n)} (\min_\theta g(c))$, because the children nodes $c \in \mathsf{ch}(n)$ have disjoint scopes due to decomposability and thus do not share any parameters.

Since the time to solve the linear program at each sum node depends only on the number of children of the sum node, which is bounded, we consider this time constant when calculating the runtime of the full algorithm. Thus, we can compute $\mathsf{CW}_p$ and the corresponding transport plan between two circuits in time linear in the size of the coupling circuit, or equivalently, quadratic in the size of the original input circuits. Appendix C.6 presents correctness proof of the algorithm in more detail.

### 3.3 RELATION TO OPTIMAL TRANSPORT FOR GMMS

As probabilistic circuits with Gaussian input distributions can be interpreted as deep, compact representations of Gaussian mixture models (GMMs), existing works studying optimal transport for GMMs (Chen et al., 2018; Delon & Desolneux, 2020) are highly relevant. In particular, our proposed notion of Circuit Wasserstein distance is closely related to the Mixture Wasserstein distance ($\mathsf{MW}_2$) introduced by Delon & Desolneux (2020), who also derived an upper bound on the true Wasserstein distance by restricting the coupling set to a GMM structure with quadratic number of components and computed this metric by solving a linear program.

We can in fact directly leverage this algorithm to compute a bound on 2-Wasserstein distance between PCs. Specifically, we can "unroll" PCs with Gaussian inputs into their shallow representations which correspond to GMMs and them compute $\mathsf{MW}_2$ between those unrolled GMMs. However, the shallow representation of a PC may be exponentially larger than the size of the original circuit, making this naive approach intractable; nevertheless, we consider this approach as a baseline for our proposed approach and provide a detailed runtime comparison in Section 5. Furthermore, we observe that $\mathsf{MW}_p$ will be no larger than our proposed $\mathsf{CW}_p$ because a coupling circuit can also be unrolled into a GMM and thus must be in the coupling set for $\mathsf{MW}_p$; we also empirically compare the efficacy of these two metrics in bounding the true Wasserstein distance in Section 5.

## 4 PARAMETER LEARNING OF PCS USING WASSERSTEIN DISTANCE

Motivated by past works that train generative models by minimizing the Wasserstein distance between the model and the empirical data distribution (Rout et al., 2022; Salimans et al., 2018; Tolstikhin et al., 2018; Arjovsky et al., 2017), we investigate the applicability of minimizing the Wasserstein distance between a PC and data as a means of learning the parameters of a given PC structure.

Formally, suppose we have a dataset $\mathcal{D} = \{\mathbf{y}^{(k)}\}_{k=1}^n$ that induces the empirical probability measure $\hat{Q}$. Then for a given PC structure, we find its parameters $\theta$ to optimize the following:

$$\min_\theta \mathsf{W}_p^p(P_\theta, \hat{Q}) = \min_\theta \inf_{\gamma \in \Gamma(P_\theta, \hat{Q})} \mathbb{E}_{\gamma(\mathbf{x}, \mathbf{y})}[\|\mathbf{x} - \mathbf{y}\|_p^p]$$

$$= \min_\theta \inf_{\gamma \in \Gamma(P_\theta, \hat{Q})} \frac{1}{n} \sum_{k=1}^n \mathbb{E}_{\gamma(\mathbf{x}|\mathbf{y}^{(k)})} \left[ \left\| \mathbf{x} - \mathbf{y}^{(k)} \right\|_p^p \right]. \tag{4}$$

Note that the second line in Equation 4 comes from rewriting $\gamma(\mathbf{x}, \mathbf{y}) = \gamma(\mathbf{x}|\mathbf{y})\gamma(\mathbf{y})$, then applying linearity of integration since $\hat{Q}$ is an empirical distribution. Unfortunately, solving the above optimization problem is computationally hard.

**Theorem 2.** Suppose $P_\theta$ is a smooth and decomposable probabilistic circuit, and $\hat{Q}$ is an empirical distribution induced by a dataset $\mathcal{D} = \{\mathbf{y}^{(k)}\}_{k=1}^n$. Then computing the parameters $\theta$ that minimizes the empirical Wasserstein distance $\mathsf{W}_p^p(P_\theta, \hat{Q})$ (i.e., solving Equation 4) is NP-hard.

We can show the above by a reduction from $k$-means clustering (Appendix C.5).

### 4.1 WASSERSTEIN-MINIMIZATION: AN ITERATIVE ALGORITHM

We again tackle this computational hardness by imposing a circuit structure on the coupling measure, allowing us compute the Wasserstein objective and optimize it efficiently.

**Definition 5** (Empirical Circuit Wasserstein distance). Let $P$ be a PC distribution and $\hat{Q}$ an empirical distribution induced by a dataset $\mathcal{D} = \{\mathbf{y}^{(k)}\}_{k=1}^n$. The $p$-Empirical Circuit Wasserstein distance between $P$ and $\hat{Q}$ is

$$\mathsf{ECW}_p^p(P, \hat{Q}) = \min_{\gamma} \frac{1}{n} \sum_{k=1}^n \mathbb{E}_{\gamma(\mathbf{x}|\mathbf{y}^{(k)})} \left[ \left\| \mathbf{x} - \mathbf{y}^{(k)} \right\|_p^p \right],$$

where $\gamma(\mathbf{x}, \mathbf{y} = \mathbf{y}^{(k)})$ satisfies the following: (i) for each $k \in \{1, \ldots, n\}$, $\gamma(., \mathbf{y} = \mathbf{y}^{(k)})$ is a PC with the same structure as $P$ (but not necessarily the same parameters) that normalizes to $1/n$, and (ii) $\sum_{k=1}^n \gamma(\mathbf{x}, \mathbf{y} = \mathbf{y}^{(k)}) = P(\mathbf{x})$.

A coupling satisfying the above structure clearly satisfies the marginal constraints and is in $\Gamma(P, \hat{Q})$. Therefore, the empirical Circuit Wasserstein distance upper-bounds the empirical Wasserstein distance: $\mathsf{W}_p(P, \hat{Q}) \leq \mathsf{ECW}_p(P, \hat{Q})$. We will thus learn the parameters of PCs by minimizing this upper bound, which can be computed efficiently as we show next.

We now present our *iterative algorithm* for minimum-Wasserstein parameter learning. In particular, we wish to learn the circuit parameters $\theta$ that minimizes $\mathsf{ECW}_p^p(P_\theta, \hat{Q})$ which is in turn a minimization problem over couplings $\gamma$. Thus, we alternate between (i) optimizing the coupling given the current circuit parameters and (ii) updating the circuit parameters given the current coupling.

Let us first discuss step (i) which computes $\mathsf{ECW}_p^p(P_\theta, \hat{Q})$ for a given $\theta$ and in the process find the corresponding coupling $\gamma$. First, rather than materializing a PC to represent $\gamma(., \mathbf{y} = \mathbf{y}^{(k)})$ for each $k$ as described in Definition 5, we can equivalently model a single coupling circuit $\gamma$ as having the same structure as $P$ and a set of parameters $\{w_{r,c,k}\}_{k=1}^n$ for each parameter $\theta_{r,c}$ in $P$. Then optimizing the coupling circuit parameters amounts to minimizing the expected distance according to the coupling distribution, similar to computing $\mathsf{CW}$, and can be done efficiently by solving a small linear program at each sum node. Here, we have the following marginal-matching constraints: $\sum_{k=1}^n w_{r,c,k} = \theta_{r,c}$ for each sum node $r$ and child $c$ and $\sum_c w_{r,c,k} = 1/n$ for each $k$.

Interestingly, the above linear program at each sum node is a variation of the continuous knapsack problem (Michael Goodrich, 2002) and thus has a closed-form solution. In particular, the solution results in a coupling circuit with each weight $w_{c,k}$ being either $\frac{1}{n}$ or zero (details in Appendix C.7). Intuitively, the coupling circuit parameters $w$ describe how each data point is routed through the circuit; because the optimal coupling is deterministic—each data point is either routed wholly through an edge or not at all—we obtain a *transport plan* between the learned PC and empirical distribution.

Next, we discuss step (ii) which estimates the parameters $\theta$ of PC $P$ from a given coupling $\gamma$. Because the coupling has the same structure as $P$, and its weights $\{w_{r,c,k}\}$ satisfy marginal-matching constraints, we can simply extract the PC parameters: $\theta_{r,c} = \sum_{k=1}^n w_{r,c,k}$.

The above two steps are repeated iteratively until convergence; a pseudocode for the complete algorithm is provided in Appendix B). Due to the closed-form solution of the LP, the time complexity of one iteration of our algorithm is linear in both the size of the circuit and the size of the dataset, and our algorithm is also guaranteed to converge (potentially to a local minimum) as every iteration only decreases or preserves the empirical Wasserstein objective (Appendix C.8). Nevertheless, finding the global optimum parameters minimizing the Wasserstein distance is still NP-hard, and our proposed efficient algorithm may get stuck at a local minimum, similar to existing maximum-likelihood parameter learning approaches.

We observe interesting parallels between our proposed Wasserstein-Minimization (WM) method and Expectation-Maximization (EM) for maximum-likelihood parameter learning. EM is an iterative algorithm that alternates between (i) computing the expected likelihood (marginalizing out the latent variables) of current parameters in the E-step and (ii) estimating the parameters that maximize this in the M-step, which is analogous to the two steps of WM: (i) computing the ECW for current parameters and (ii) updating the parameters to minimize ECW.

## 5   EXPERIMENTS

In this section, we first empirically evaluate our proposed algorithm for computing $\mathsf{CW}_p$ against the algorithm proposed by Delon & Desolneux (2020) for computing $\mathsf{MW}_2$. We then compare

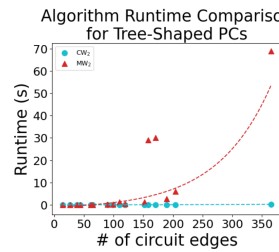 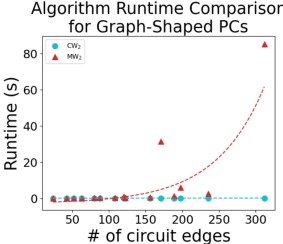

Figure 2: Runtime of Wasserstein-type distance computation using our approach (blue dots) and the baseline (red triangles). Each data point represents a pair of circuit structures corresponding to a fixed circuit branching factor and fixed number of random variables in the circuit scope, averaged over 100 random parameter initializations. For circuits larger than those depicted, the baseline approach runs out of memory. See Appendix 5.1 for runtime results of our approach on larger circuits.

the density-estimation capabilities of circuits learned using our proposed Wasserstein-Minimzation (WM) algorithm against the Expectation-Maximization (EM) algorithm for PCs. Specifically, we aim to answer the following three questions:

1. How does the runtime of our algorithm for computing $CW_2$ scale with the size of the circuit in practice, and how does that compare to $MW_2$ computation (Delon & Desolneux, 2020)?

2. How close is the computed value of $CW_2$ to $MW_2$?

3. How do circuits learned using our Wasserstein-Minimzation algorithm compare with circuits learned using the classic Expectation-Maximization algorithm?

## 5.1 RUNTIME EXPERIMENTS FOR COMPUTING $CW_2$

To evaluate the runtime of computing $CW_2$, we consider a fixed variable scope and randomly construct a balanced hierarchical scope partitioning. Then, we randomly construct two PCs with this partitioning such that they are compatible; the PCs are constructed with a fixed sum node branching factor and fixed rejoin probability—i.e., the chance that a graph connection to an existing node in the PC will be made to add a child rather than creating a new node for the child, which is 0% in the case of trees and set to 50% in the case of graphs. We implement our algorithm as detailed in Algorithm 1 to compute the optimal transport map and value for $CW_2$, as well as also implement a PC-to-GMM unrolling algorithm and the algorithm proposed by Chen et al. (2018) to compute $MW_2$ (Delon & Desolneux, 2020).[3]

The results are summarized in Figure 2, which demonstrate the quadratic runtime of our algorithm in the size of the original circuits, which sharply contrasts with the exponential runtime of the naive computation of $MW_2$ by circuit unrolling. For circuits with just over one hundred edges (after which the algorithm runs out of memory), naively computing $MW_2$ is over *three hundred times slower* than computing $CW_2$ with our feedforward algorithm. We include runtime results for circuits two orders of magnitude larger in Appendix D.1.

## 5.2 COMPARING THE QUANTITIES OF $CW_2$ AND $MW_2$

Because both $CW_2$ and $MW_2$ upper-bound the true 2-Wasserstein distance, the smaller the values, the tighter the bound. We adopt the same framework as we did for runtime experiments to randomly construct compatible PCs and compute $CW_2$ and $MW_2$ between them. Due to the exponential blowup of computing $MW_2$ it quickly becomes impractical to compute (see Section 5.1); however, we still attempt to provide some empirical insight into the difference between $CW_p$ and $MW_p$. Empirically, the difference between $CW_p$ and $MW_p$ grows with circuit size (see Appendix D.4 for detailed figures).

---

[3]Code for our implementation and experiments will be made available online upon acceptance of the paper.

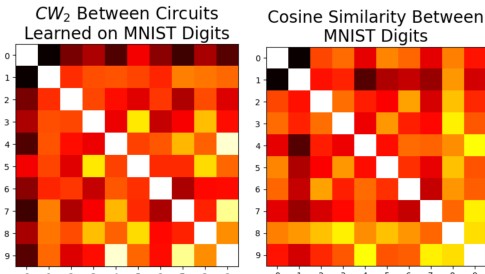

Figure 3: *(Left)* $\mathsf{CW}_2$ between circuits learned on the given dataset partitions corresponding to images with only that digit; lighter pixels represent a lower $\mathsf{CW}_p$. *(Right)* Cosine similarity between dataset partitions; lighter pixels represent a higher cosine similarity.

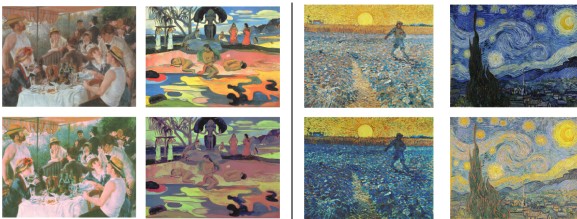

Figure 4: Color transfer between images using coupling circuits. In each of the above figures, the top two images represent images $A$ and $B$, and the bottom two images represent image $A$ transferred to image $B$'s color palette and image $B$ transferred to image $A$'s color palette respectively.

We also note that while the ratio $\frac{\mathsf{CW}_2}{\mathsf{MW}_2}$ is lower for circuits of higher depth, this can be attributed to higher-depth circuits potentially having fewer learnable parameters (and thus less opportunity for error to compound) relative to their size. A circuit with a large scope size (and thus high depth) but small sum node branching factor can have the same number of edges but far fewer parameters than a shallow circuit with a large sum node branching factor.

## 5.3 Computing $\mathsf{CW}_2$ Between Learned Circuits

To support the feasibility of our algorithm when applied to large, high-dimensional PCs, we computed $\mathsf{CW}_p$ between circuits learned on the MNIST dataset (LeCun et al., 1998) – a 784-dimensional handwritten digits image dataset. Specifically, we first partitioned the dataset into 10 classes by the digit depicted in the image, and then learned one circuit per class using the HCLT structure learning algorithm (Liu & Van den Broeck, 2021) with a fixed block size of 4 (resulting in each circuit having over 11k edges) and Expectation-Maximization parameter learning algorithm (Desana & Schnörr, 2016) implemented in PyJuice (Liu et al., 2024). We then computed $\mathsf{CW}_2$ between each pair of circuits (which took under two seconds per pair with our implementation), and plot these values along with the average cosine similarity between the classes in Figure 3. We found that cosine similarity between classes was inversely correlated with $\mathsf{CW}_p$ between circuits learned on those classes with a correlation coefficient of $r = -0.78$, supporting the utility of our proposed distance metric.

## 5.4 Color Transfer Between Images Using Optimal Transport

We adopt an application of optimal transport shown by $Delon\&Desolneux$ (2020), whereby we transport the *color histogram*—the 3-dimensional probability distribution of pixel color values—of image $a$ to that of another image $b$. To do this, we learn compatible PCs $P(\mathbf{X})$ and $Q(\mathbf{Y})$ over the color distributions of images $a$ and $b$, compute the optimal coupling circuit $C(\mathbf{X}, \mathbf{Y})$, and transport each pixel with color value $\mathbf{x}$ to the corresponding pixel $\mathbb{E}_C[\mathbf{Y}|\mathbf{X} = \mathbf{x}]$ (which can be computed tractably). See Figure 4 for two examples.

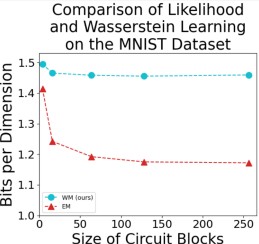

Figure 5: Visualization of the performance of PCs learned using Expectation Maximization (red triangles) and Wasserstein Minimization (our approach, blue dots). The bits-per-dimension (bpd) of the learned circuits does not decrease significantly with an increase in circuit size for circuits learned using the empirical Wasserstein distance.

## 5.5 WASSERSTEIN-MINIMIZATION FOR CIRCUIT PARAMETER LEARNING

To determine the performance of our proposed Wasserstein-Minimization algorithm on density estimation tasks, we consider learning the parameters of circuits of various sizes from the MNIST benchmark dataset (LeCun et al., 1998). We first generated the structure of the circuits using the HCLT algorithm (Liu & Van den Broeck, 2021) implementation provided in PyJuice (Liu et al., 2024), varying the "block size" to increase or decrease the number of parameters. We then learned two sets of circuit parameters per structure per block size: one set of parameters was learned using mini-batch EM (Desana & Schnörr, 2016), and the other set was learned using our proposed WM algorithm. We performed early stopping for the EM algorithm that stops training once the point of diminishing returns has been surpassed. All experiments were ran on a single NVIDIA L40s GPU.

In terms of bits-per-dimension, we observe that our algorithm performs nearly as well as EM for small circuits (block size 4). However, as the size of the circuit increases, the performance of our algorithm quickly stagnates; empirically, our WM approach does not seem to take full advantage of the larger parameter space of larger models, with models orders of magnitude larger having better but still comparable performance to their smaller counterparts. We refer to Figure 5 for more details.

## 6 CONCLUSION

This paper studied the optimal transport problem for probabilistic circuits. We introduced a Wasserstein-type distance $\mathsf{CW}_p$ between two PCs an proposed an efficient algorithm that computes the distance and corresponding optimal transport plan in quadratic time in the size of the input circuits, provided that their circuit structures are compatible. We show that $\mathsf{CW}_p$ always upper-bounds the true Wasserstein distance, and that—when compared to the naive application of an existing algorithm for computing a Wasserstein-type distance between GMMs to PCs—the former is exponentially faster to compute between circuits. Lastly, we propose an iterative algorithm to minimize the empirical Wasserstein distance between a circuit and data, suggesting an alternative, viable approach to parameter estimation for PCs which is mainly done using maximum-likelihood estimation. While performance was competitive with the EM algorithm for small circuits, we leave as future work to get Wasserstein Minimization to fully exploit the increased expressiveness of larger models.

We consider this work an initial stepping stone towards a deeper understanding of optimal transport theory for probabilistic circuits. Future work includes exploring more expressive formulations of coupling circuits to obtain a tighter bound on Wasserstein distance—such as relaxing the node-by-node parameter constraints to only require that the whole circuit matches marginal distributions to the original circuits. Our work also leaves open the possibility of extending the marginal-preserving properties of coupling circuits to the multimarginal setting for multimarginal generative modeling with PCs, and computing Wasserstein barycenters for PCs. Moreover, we envision that the tractable computation of a Wasserstein-type distance and transport plan between expressive models such as PCs can lead to further development in various Wasserstein-based machine learning approaches.

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

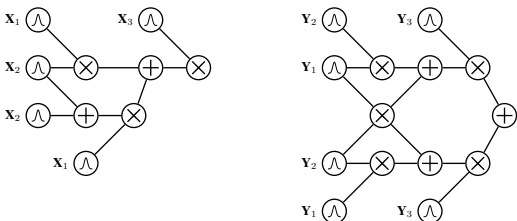

Figure 6: Two compatible circuits over $\mathbf{X} = (X_1, X_2, X_3)$ and $\mathbf{Y} = (Y_1, Y_2, Y_3)$. Note that compatibility does not require the same (or even notably similar) structures apart from the hierarchical scope partitioning.

## A  ADDITIONAL FIGURES

We include an example of compatible circuits with different structures in Figure 6.

## B  ALGORITHM FOR MINIMUM WASSERSTEIN PARAMETER ESTIMATION

Our proposed algorithm is broadly divided into two steps: an inference step and a minimization step. These steps are performed iteratively until model convergence. The inference step populates a cache, which stores the expected distance of each data point at each node in the circuit. This inference step is done in linear time in a bottom-up recursive fashion, making use of the cache for already-computed results. This is provided in algorithm 2.

The minimization step is done top-down recursively, and seeks to route the data at a node to its children in a way that minimizes the total expected distance between the routed data at each child and the sub-circuit. The root node is initialized with all data routed to it. At a sum node, each data point is routed to the child that has the smallest expected distance to it (making use of the cache from the inference step), and the edge weight corresponding to a child is equal to the proportion of data routed to that child; at a product node, the data point is routed to both children. Input node parameters are updated to reflect the empirical distribution of the data routed to that node. The minimization step is thus also done in linear time, and we note that this algorithm guarantees a non-decreasing objective function (see Appendix C.8 for a proof). The algorithm for this is provided in algorithm 3.

---

**Algorithm 2** INFERENCE$(n, D)$: returns a cache storing the distance between each data point $d_j \in D$ and each sub-circuit rooted at $n$, where $n$ has children $c_i$. For conciseness, we omit checking for cache hits

---

    **for** $c_i \in n.\text{children}$ **do**
        INFERENCE$(c_i, D)$                                   ▷ recursively build cache
    **if** $n$ is a product node **then**
        **for** $d_j \in D$ **do**
            cache$[n, d_j] \leftarrow \sum_i$cache$[c_i, d_j]$
    **if** $n$ is a sum node **then**
        **for** $d_j \in D$ **do**
            cache$[n, d_j] \leftarrow \sum_i \theta_i$cache$[c_i, d_j]$
    **if** $n$ is an input node **then**
        **for** $d_j \in D$ **do**
            cache$[n, d_j] \leftarrow dist(n, d_j)$ ▷ here, $dist(n, d_j)$ is the expected distance between $n$ and $d_j$
    **return** cache

---

---

**Algorithm 3** LEARN($n, D$, cache): learns the parameters of circuit rooted at $n$ on data points $d_j \in D$

---

**if** not all parents of $n$ have been learned **then**
    **return**               ▷ We only call this method on nodes who's parents have all been learned
**if** $n$ is a product node **then**
    **for** $c_i \in n$.children **do**
        routing$[c_i] \leftarrow$ routing$[n]$               ▷ products route their data to their children
**if** $n$ is a sum node **then**
    $\forall \theta_i, \theta_i \leftarrow 0$                              ▷ zero out parameters
    **for** $d_j \in$ routing$[n]$ **do**         ▷ route data points at current node to children
        $c_i \leftarrow \arg\min_{c_i}$ cache$[c_i, d_j]$     ▷ $c_i$ is the child node of $n$ for which $d_j$ has the lowest distance
        routing$[c_i] \leftarrow d_j$                         ▷ route $d_j$ to $c_i$
        $\theta_i \leftarrow \theta_i + \frac{1}{|\text{routing}[n]|}$        ▷ update parameter weight
**if** $n$ is an input node **then**
    $n$.parameters $\leftarrow$ parameters matching empirical distribution of routing$[n]$

---

## C   PROOFS

### C.1   HARDNESS PROOF OF THE $\infty$-WASSERSTEIN DISTANCE BETWEEN CIRCUITS

**Theorem 1.** Suppose $P$ and $Q$ are probabilistic circuits over $n$ Boolean variables. Then computing the $\infty$-Wasserstein distance between $P$ and $Q$ is coNP-hard, even when $P$ and $Q$ are *deterministic* and *structured-decomposable*.

*Proof.* We will prove hardness by reducing the problem of deciding equivalence of two DNF formulas, which is known to be coNP-hard, to Wasserstein distance computation between two compatible PCs.

Consider a DNF $\alpha$ containing $m$ terms $\{\alpha_1, \ldots, \alpha_m\}$ over Boolean variables $\mathbf{X}$. We will construct a PC $P_\alpha$ associated with this DNF as follows. For each term $\alpha_i$, we construct a product of input nodes—one for each $X \in \mathbf{X}$ whose literal appears in term $\alpha_i$, $\mathbb{1}[X = 1]$ for a positive literal and $\mathbb{1}[X = 0]$ for negative. Then we construct a sum unit with uniform parameters over these products as the root of our PC: $P_\alpha = \sum_{i=1}^m \frac{1}{m} P_{\alpha_i}$. We can easily smooth this PC by additionally multiplying $P_{\alpha_i}$ with a sum node $\frac{1}{2}\mathbb{1}[X = 0] + \frac{1}{2}\mathbb{1}[X = 1]$ for each variable $X$ that does not appear in $\alpha_i$. Furthermore, note that every product node in this circuit fully factorizes the variables $\mathbf{X}$, and thus the PC is trivially compatible with any decomposable circuit over $\mathbf{X}$ and in particular with any other PC for a DNF over $\mathbf{X}$, constructed as above.

Clearly, the above PC $P_\alpha$ assigns probability mass only to the models of $\alpha$. In other words, for any $\mathbf{x} \in \{0, 1\}^n$, $P_\alpha(\mathbf{x}) > 0$ iff $\mathbf{x} \models \alpha$ (i.e. there is a term $\alpha_i$ that is satisfied by $\mathbf{x}$). $\qquad\square$

### C.2   PROOF OF THE MARGINAL-MATCHING PROPERTIES OF COUPLING CIRCUITS

**Proposition 3.** Let $P$ and $Q$ be compatible PCs. Then any feasible coupling circuit $C$ as defined in Def. 3 matches marginals to $P$ and $Q$.

*Proof.* We will prove this by induction. Our base case is two corresponding input nodes $n_1, n_2 \in P, Q$. The sub-circuit in $C$ rooted at the product of $n_1$ and $n_2$ is a product node with copies of $n_1$ and $n_2$ as children, which clearly matches marginals to $n_1$ and $n_2$.

Now, let $n_1$ and $n_2$ be arbitrary corresponding nodes in $P$ and $Q$, and assume that the product circuits for all children of the two nodes match marginals. We then have two cases:

**Case 1: $n_1, n_2$ are product nodes**    Since the circuits are compatible, we know that $n_1$ and $n_2$ have the same number of children—let the number of children be $k$. Thus, let $c_{1,i}$ represent the $i$'th child of $n_1$, and let $c_{2,i}$ represent the $i$'th child of $n_2$. The coupling circuit of $n_1$ and $n_2$ (denoted $n$) is

a product node with $k$ children, where the $i$'th child is the coupling circuit of $c_{1,i}$ and $c_{2,i}$ (denoted $c_i$).

By induction, the distribution $P_{c_i}(\mathbf{X}, \mathbf{Y})$ at each child coupling sub-circuit matches marginals to the original sub-circuits: $P_{c_i}(\mathbf{X}) = P_{c_{1,i}}(\mathbf{X})$, and $P_{c_i}(\mathbf{Y}) = P_{c_{2,i}}(\mathbf{Y})$. $n_1$ and $n_2$ being product nodes means that $P_{n_1}(\mathbf{X}) = \prod_i P_{c_{1,i}}(\mathbf{X})$ and $P_{n_2}(\mathbf{Y}) = \prod_i P_{c_{2,i}}(\mathbf{Y})$, so thus $P_n(\mathbf{X}) = \prod_i P_{c_i}(\mathbf{X}) = \prod_i P_{c_{1,i}}(\mathbf{X})$ and $P_n(\mathbf{Y}) = \prod_i P_{c_i}(\mathbf{Y}) = \prod_i P_{c_{2,i}}(\mathbf{Y})$. Therefore, $n$ matches marginals to $n_1$ and $n_2$.

**Case 2: $n_1, n_2$ are sum nodes** Let the number of children of $n_1$ be $k_1$ and the number of children of $n_2$ be $k_2$. Let $c_{1,i}$ represent the $i$'th child of $n_1$, and let $c_{2,j}$ represent the $j$'th child of $n_2$. The coupling circuit of $n_1$ and $n_2$ (denoted $n$) is a sum node with $k_1 * k_2$ children, where the $(i,j)$'th child is the coupling circuit of $c_{1,i}$ and $c_{2,j}$ (denoted $c_{i,j}$).

By induction, the distribution $P_{c_{i,j}}(\mathbf{X}, \mathbf{Y})$ at each child coupling sub-circuit matches marginals to the original sub-circuits: $P_{c_{i,j}}(\mathbf{X}) = P_{c_{1,i}}(\mathbf{X})$, and $P_{c_{i,j}}(\mathbf{Y}) = P_{c_{2,j}}(\mathbf{Y})$. $n_1$ and $n_2$ being sum nodes means that $P_{n_1}(\mathbf{X}) = \sum_i \theta_i P_{c_{1,i}}(\mathbf{X})$ and $P_{n_2}(\mathbf{Y}) = \sum_j \theta_j P_{c_{2,j}}(\mathbf{Y})$, so thus

$$P_n(\mathbf{X}) = \sum_i \sum_j \theta_{i,j} P_{c_{i,j}}(\mathbf{X}) = \sum_i \sum_j \theta_{i,j} P_{c_{1,i}}(\mathbf{X}) = \sum_i \theta_i P_{c_{1,i}}(\mathbf{X}) = P_{n_1}(\mathbf{X})$$

$$P_n(\mathbf{Y}) = \sum_i \sum_j \theta_{i,j} P_{c_{i,j}}(\mathbf{Y}) = \sum_i \sum_j \theta_{i,j} P_{c_{2,j}}(\mathbf{Y}) = \sum_j \theta_j P_{c_{2,j}}(\mathbf{Y}) = P_{n_2}(\mathbf{Y}) \quad (5)$$

Note that we rewrite $\sum_i \theta_{i,j} = \theta_j$ and $\sum_j \theta_{i,j} = \theta_i$ by the constraints on coupling circuits. Therefore, $n$ satisfies marginal constraints. $\square$

### C.3 Proof of the Metric Properties of $\mathsf{CW}_p$

**Proposition 1** (Metric Properties of $\mathsf{CW}_p$). For any set $\mathcal{C}$ of compatible circuits, $\mathsf{CW}_p$ defines a metric on $\mathcal{C}$.

*Proof.* It is clear that $\mathsf{CW}_p$ is symmetric since construction of the coupling circuit is symmetric. Furthermore, since $\mathsf{CW}_p$ upper-bounds $\mathsf{W}_p$, it must also be non-negative.

If $\mathsf{CW}_p(P, Q) = 0$, then $\mathsf{W}_p(P, Q) = 0$ so $P = Q$. Any constraint-satisfying assignment of the parameters of a coupling circuit between $P$ and $P$ would also result in the Wasserstein objective at the root node being $0$, since the base-case computation of $\mathsf{W}_p$ at the leaf nodes would always be zero.

Now, we show that $\mathsf{CW}_p$ satisfies the triangle inequality. Let $P, Q, R \in \mathcal{C}$ be compatible PCs over random variables $\mathbf{X}, \mathbf{Y}$, and $\mathbf{Z}$, and let $d_1 = \mathsf{CW}_p(P, Q)$, $d_2 = \mathsf{CW}_p(P, R)$, and $d_3 = \mathsf{CW}_p(R, Q)$ with optimal coupling circuits $C_1, C_2$, and $C_3$. We can construct circuits $C_2(\mathbf{x}|\mathbf{z})$ and $C_3(\mathbf{y}|\mathbf{z})$ that are still compatible with $C_2$ and $C_3$, since conditioning a circuit preserves the structure. Because all of these are compatible, we can then construct circuit $C(\mathbf{X}, \mathbf{Y}, \mathbf{Z}) = C_2(\mathbf{X}|\mathbf{Z})C_3(\mathbf{Y}|\mathbf{Z})R(\mathbf{Z})$. Thus, $C$ is a coupling circuit of $P, Q$, and $R$ such that $C_2(\mathbf{x}, \mathbf{y}) = \int C(\mathbf{x}, \mathbf{y}, \mathbf{z})d\mathbf{z}$ and $C_3(\mathbf{y}, \mathbf{z}) = \int C(\mathbf{x}, \mathbf{y}, \mathbf{z})d\mathbf{x}$. Then we have:

$$\mathsf{CW}_p(P, Q) = \int \|\mathbf{x} - \mathbf{y}\|_p^p C_1(\mathbf{x}, \mathbf{y})d\mathbf{x}d\mathbf{y} = \int \|(\mathbf{x} - \mathbf{z}) - (\mathbf{y} - \mathbf{z})\|_p^p C(\mathbf{x}, \mathbf{y}, \mathbf{z})d\mathbf{x}d\mathbf{y}d\mathbf{z}$$

$$\leq \int \|\mathbf{x} - \mathbf{z}\|_p^p C_2(\mathbf{x}, \mathbf{z})d\mathbf{x}d\mathbf{z} + \int \|\mathbf{z} - \mathbf{y}\|_p^p C_3(\mathbf{y}, \mathbf{z})d\mathbf{y}d\mathbf{z}$$

$$= \mathsf{CW}_p(P, R) + \mathsf{CW}_p(R, Q)$$

Thus, $\mathsf{CW}_p$ satisfies the triangle inequality, which concludes the proof. $\square$

### C.4 Recursive Computation of the Wasserstein Objective

Referring to Definition 4, the Wasserstein objective for a given coupling circuit $C(\mathbf{x}, \mathbf{y})$ is the expected distance between $\mathbf{x}$ and $\mathbf{y}$. Below, we demonstrate that the Wasserstein objective at a sum

node that decomposes into $C(\mathbf{x}, \mathbf{y}) = \sum_i \theta_i C_i(\mathbf{x}, \mathbf{y})$ is simply the weighted sum of the Wasserstein objectives at its children:

$$\mathbb{E}_{C(\mathbf{x},\mathbf{y})}[\|\mathbf{x} - \mathbf{y}\|_p^p] = \int \|\mathbf{x} - \mathbf{y}\|_p^p C(\mathbf{x}, \mathbf{y}) d\mathbf{x} d\mathbf{y} = \int \|\mathbf{x} - \mathbf{y}\|_p^p \sum_i \theta_i C_i(\mathbf{x}, \mathbf{y}) d\mathbf{x} d\mathbf{y}$$

$$= \sum_i \theta_i \int \|\mathbf{x} - \mathbf{y}\|_p^p C_i(\mathbf{x}, \mathbf{y}) d\mathbf{x} d\mathbf{y} = \sum_i \theta_i \, \mathbb{E}_{C_i(\mathbf{x},\mathbf{y})}[\|\mathbf{x} - \mathbf{y}\|_p^p] \quad (6)$$

Now, consider a decomposable product node, where $C(\mathbf{x}, \mathbf{y}) = C_1(\mathbf{x}_1, \mathbf{y}_1) C_2(\mathbf{x}_2, \mathbf{y}_2)$ [4]. Below, we see that the Wasserstein objective at the parent is simply the *sum* of the Wasserstein objectives at its children:

$$\mathbb{E}_{C(\mathbf{x},\mathbf{y})}[\|\mathbf{x} - \mathbf{y}\|_p^p] = \int \|\mathbf{x} - \mathbf{y}\|_p^p C(\mathbf{x}, \mathbf{y}) d\mathbf{x} d\mathbf{y} = \int \|\mathbf{x} - \mathbf{y}\|_p^p C_1(\mathbf{x}_1, \mathbf{y}_1) C_2(\mathbf{x}_2, \mathbf{y}_2) d\mathbf{x} d\mathbf{y}$$

$$= \int (\|\mathbf{x}_1 - \mathbf{y}_1\|_p^p + \|\mathbf{x}_2 - \mathbf{y}_2\|_p^p) \times C_1(\mathbf{x}_1, \mathbf{y}_1) C_2(\mathbf{x}_2, \mathbf{y}_2) d\mathbf{x}_1 d\mathbf{x}_2 d\mathbf{y}_1 d\mathbf{y}_2$$

$$= \left( \int \|\mathbf{x}_1 - \mathbf{y}_1\|_p^p C_1(\mathbf{x}_1, \mathbf{y}_1) d\mathbf{x}_1 d\mathbf{y}_1 \right) + \left( \int \|\mathbf{x}_2 - \mathbf{y}_2\|_p^p) C_2(\mathbf{x}_2, \mathbf{y}_2) d\mathbf{x}_2 d\mathbf{y}_2 \right)$$

$$= \mathbb{E}_{C_1(\mathbf{x}_1,\mathbf{y}_1)}[\|\mathbf{x}_1 - \mathbf{y}_1\|_p^p] + \mathbb{E}_{C_2(\mathbf{x}_2,\mathbf{y}_2)}[\|\mathbf{x}_2 - \mathbf{y}_2\|_p^p] \quad (7)$$

Thus, we can push computation of Wasserstein objective down to the leaf nodes of a coupling circuit.

## C.5 HARDNESS PROOF OF COMPUTING MINIMUM-WASSERSTEIN PARAMETERS

**Theorem 2.** Suppose $P_\theta$ is a smooth and decomposable probabilistic circuit, and $\hat{Q}$ is an empirical distribution induced by a dataset $\mathcal{D} = \{\mathbf{y}^{(k)}\}_{k=1}^n$. Then computing the parameters $\theta$ that minimizes the empirical Wasserstein distance $\mathsf{W}_p^p(P_\theta, \hat{Q})$ (i.e., solving Equation 4) is NP-hard.

*Proof.* We will prove this hardness result by reducing $k$-means clustering—which is known to be NP-hard (Dasgupta, 2008)—to learning the minimum Wasserstein parameters of a circuit. Consider a set of points $x_1 ... x_n \in \mathbb{R}^d$ and a number of clusters $k$. We will construct a Gaussian PC $C$ associated with this problem as follows: the root of $C$ is a sum node with $k$ children; each child is a product node with $d$ univariate Gaussian input node children (so each product node is a multivariate Gaussian comprised of independent univariate Gaussians). Minimizing the parameters of $C$ over $x_i$ corresponds to finding a routing of data points $x_i$ that minimizes the total distance between all $x_i$'s and the mean of the multivariate Gaussian child each $x_i$ is routed to. A solution to $k$-means can be retrieved by taking the mean of each child of the root sum node to be the center of each of $k$ clusters. $\square$

## C.6 PROOF OF THE OPTIMALITY OF COUPLING CIRCUIT PARAMETER LEARNING IN ALGORITHM 1

**Proposition 4.** Suppose $P$ and $Q$ are compatible probabilistic circuits with coupling circuit $C$. Then the parameters of $C$—and thus $\mathsf{CW}_p$—can be computed exactly in a bottom-up recursive fashion.

*Proof.* We will construct a recursive argument showing that the optimal parameters of $C$ can be computed exactly. Let $n \in C$ be some non-input node in the coupling circuit $C$ that is the product of nodes $n_1$ and $n_2$ in $P$ and $Q$ respectively. Then we have three cases:

**Case 1: $n$ is a product node with input node children** Due to the construction of the coupling circuit, $n$ must have two children that are input nodes with scopes $\mathbf{X}_k$ and $\mathbf{Y}_k$. Thus, $\mathsf{CW}_p(n)$ is simply computed in closed-form as the $p$-Wasserstein distance between the input distributions.

---

[4]We assume for notational simplicity that product nodes have two children, but it is straightforward to rewrite a product node with more than two children as a chain of product nodes with two children each and see that our result still holds.

**Case 2:** $n$ **is a product node with non-input node children** By recursion, $\mathsf{CW}_p(n) = \sum_i \mathsf{CW}_p(c_i)$ for each child $c_i$ of $n$ (see 7).

**Case 3:** $n$ **is a sum node** Let $\theta_{i,j}$ be the parameter corresponding to the product of the $i$-th child of $n_1$ and $j$-th child of $n_2$. We want to solve the following optimization problem $\inf \mathbb{E}_{P_n(\mathbf{X},\mathbf{Y})}[\|\mathbf{X} - \mathbf{Y}\|_p^p]$, which can be rewritten as follows:

$$\inf \mathbb{E}_{P_n(\mathbf{X},\mathbf{Y})}[\|\mathbf{X} - \mathbf{Y}\|_p^p] = \inf \int_{\mathbb{R}^d \times \mathbb{R}^d} \|\mathbf{X} - \mathbf{Y}\|_p^p P_n(\mathbf{X},\mathbf{Y}) d\mathbf{X} d\mathbf{Y} \tag{8}$$

Rewriting the distribution of $n$ as a mixture of its child distributions $c_{i,j}$, we get:

$$= \inf_{\theta, P_{i,j}} \int_{\mathbb{R}^d \times \mathbb{R}^d} \|\mathbf{X} - \mathbf{Y}\|_p^p \sum_{i,j} \theta_{i,j} P_{c_{i,j}}(\mathbf{X},\mathbf{Y}) d\mathbf{X} d\mathbf{Y} \tag{9}$$

Due to linearity of integrals, we can bring out the sum:

$$= \inf_{\theta, P_{i,j}} \sum_{i,j} \theta_{i,j} \int_{\mathbb{R}^d \times \mathbb{R}^d} \|\mathbf{X} - \mathbf{Y}\|_p^p P_{c_{i,j}}(\mathbf{X},\mathbf{Y}) d\mathbf{X} d\mathbf{Y} \tag{10}$$

Lastly, due to the acyclicity of PCs, we can separate out $\inf_{\theta_i, P_{i,j}}$ into $\inf_{\theta_i} \inf_{P_{i,j}}$ and push the latter infimum inside the sum.

$$= \inf_{\theta} \sum_{i,j} \theta_{i,j} \left( \inf_{P_{i,j}} \int_{\mathbb{R}^d \times \mathbb{R}^d} \|\mathbf{X} - \mathbf{Y}\|_p^p P_{c_{i,j}}(\mathbf{X},\mathbf{Y}) d\mathbf{X} d\mathbf{Y} \right) \tag{11}$$

Thus, we can solve the inner optimization problem first (corresponding to the optimization problems at the children), and then the outer problem (the optimization problem at the current node). Therefore, a bottom-up recursive algorithm is exact. $\square$

## C.7 DERIVING A CLOSED-FORM SOLUTION TO THE LINEAR PROGRAMS FOR PARAMETER UPDATES

For a sum node $s$ with $m$ children $s_1...s_m$ and a dataset with $n$ data points $d_1...d_n$ each with weight $w_j$, we construct a linear program with $m * n$ variables $x_{i,j}$ as follows:

$$\min \quad \sum_{i=1}^{m} \sum_{j=1}^{n} \mathbb{E}_{s_i}[\|\mathbf{X} - d_j\|_2^2] x_{i,j} \qquad \text{s.t.} \quad \sum_{i=1}^{m} x_{i,j} = w_j \; \forall j$$

Note that the constraints do not overlap for differing values of $j$. Thus, we can break this problem up into $n$ smaller linear programs, each with the following form:

$$\min \quad \sum_{i=1}^{m} \mathbb{E}_{s_i}[\|\mathbf{X} - d_j\|_2^2] x_{i,j} \qquad \text{s.t.} \quad \sum_{i=1}^{m} x_{i,j} = w_j$$

The only constraint here requires that the sum of objective variables is equal to $w_j$. Thus, the objective is minimized when $x_{i,j}$ corresponding to the smallest coefficient takes value $w_j$ and all other variables take value 0. Thus, the solution to the original linear program can be thought of as assigning each data point to the sub-circuit that has the smallest expected distance to it.

## C.8 PROOF THAT THE WASSERSTEIN MINIMIZATION ALGORITHM HAS A MONOTONICALLY DECREASING OBJECTIVE

**Proposition 5.** For a circuit rooted at $n$ and dataset $D$ routed to it, the Wasserstein distance between the empirical distribution of $D$ and sub-circuit rooted at $n$ will not increase after an iteration of algorithm B

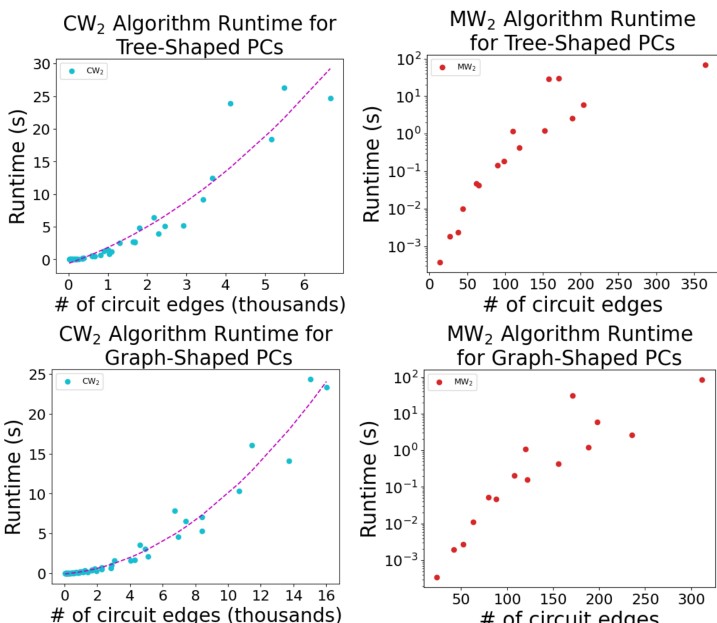

Figure 7: Runtime for algorithms computing $\mathsf{CW}_2$ and $\mathsf{MW}_2$. The first pair of graphs considers only tree-shaped PCs, whereas the second pair considers graph-shaped PCs as well. Note that the right-side graphs use logarithmic scaling. Number of circuit edges represents the number of edges in both circuits combined, and each data point represents an average over 100 runs.

*Proof.* Let $\mathbb{E}_n[D]$ denote the Wasserstein distance between the empirical distribution of $D$ and sub-circuit rooted at $n$ before an iteration of algorithm B, and let $\mathbb{E}_{n'}[D]$ denote the distance after an iteration. We will show by induction that $\mathbb{E}_{n'}[D] \leq \mathbb{E}_n[D]$. Our base case is when $n$ is an input node. By setting the parameters of $n$ to as closely match the empirical distribution of $D$ as possible, there is no parameter assignment with a lower Wasserstein distance to $D$ so thus one iteration of algorithm B does not increase the objective value.

Recursively, we have two cases:

**Case 1: $n$ is a product node** By the decomposition of the Wasserstein objective, we have that $\mathbb{E}_n[D] = \sum_i \mathbb{E}_{c_i}[D]$, which is $\geq \sum_i \mathbb{E}_{c_i'}[D] = \mathbb{E}_{n'}[D]$ by induction.

**Case 2: $n$ is a sum node** By the decomposition of the Wasserstein objective, we have that $\mathbb{E}_n[D] = \sum_i \theta_i \mathbb{E}_{c_i}[D_i]$ (where $D_i \subseteq D$ is the data routed to $n_i$), which is $\geq \sum_i \theta_i \mathbb{E}_{c_i'}[D_i] = \mathbb{E}_{n'}[D]$ by induction. Our parameter updates also update each $D_i \to D_i'$, but that also guarantees that $\mathbb{E}_{c_i'}[D_i] \geq \mathbb{E}_{c_i'}[D_i']$ since $D_i = D_i'$ is within the feasible set of updates for $D_i$. Thus, $\mathbb{E}_n[D] \geq \mathbb{E}_{n'}[D]$, so therefore the Wasserstein objective is monotonically decreasing. $\square$

## D   ADDITIONAL EXPERIMENTAL RESULTS

### D.1   RUNTIME EXPERIMENTS FOR COMPUTING $\mathsf{CW}_p$

The value obtained for each circuit size is averaged over 100 runs, and we omit data points for experiments that ran out of memory. Lastly, all experiments were ran on a machine with an Intel Core i9-10980XE CPU and 256Gb of RAM—these experiments made no use of GPUs. To solve the linear programs we used Gurobi (Gurobi Optimization, LLC, 2024), a commercial linear program solver available under academic license.

Figure 7 shows the complete set of runtime results.

| | EM Circuit | | Deterministic WM | | Stochastic WM | |
|---|---|---|---|---|---|---|
| Circuit Block Size | $W_2$ | BPD | $W_2$ | BPD | $W_2$ | BPD |
| 4 | 32631 | **1.414** | 32766 | 1.495 | **29963** | 1.532 |
| 16 | 32873 | **1.242** | 32751 | 1.465 | **29984** | 1.509 |
| 64 | 33264 | **1.192** | 32749 | 1.458 | **30999** | 1.485 |
| 128 | 33737 | **1.175** | 32749 | 1.455 | **31483** | 1.474 |
| 256 | 34974 | **1.172** | 32528 | 1.459 | **32520** | 1.459 |

Table 1: Comparison of Wasserstein objective value and bits-per-dimension (BPD) between circuits learned via EM and WM (our approach), lower is better. The lowest value for each circuit size is bolded. Deterministic WM routes all data points optimally to minimize the Wasserstein objective, while stochastic WM randomly routes a data point to one of the children with probability $p$ and optimally with probability $1 - p$.

## D.2 EMPIRICAL WASSERSTEIN PARAMETER ESTIMATION EXPERIMENTAL RESULTS

We investigated the computed Wasserstein objective and bits-per-dimension (BPD) of circuits of various sizes learned using EM and WM (our method). We found that larger circuits trained via EM have a significantly lower BPD than smaller circuits, which was not the case for circuits trained via WM. Looking at the Wasserstein objective for these circuits, we see that bpd is not directly correlated with the Wasserstein objective; circuits with a lower Wasserstein objective can have a slightly higher bpd, and vice versa.

Lastly, we consider a modification of Algorithm B that employs *stochastic routing* of data at sum nodes; succinctly, we introduce hyperparameter $p$ that introduces a probability $p$ that a given data point is routed randomly with uniform probability to any given child node, and a probability $1 - p$ that the data point is routed optimally as detailed in Algorithm B. When $p = 0$, we refer to this as *deterministic WM*; otherwise, we refer to the algorithm as *stochastic WM*.

In our experiments, we found that $p = 0.1$ yields the best results for minimizing the Wasserstein objective. For circuits of block size 4, we observe that this significantly decreases the Wasserstein objective without a significant change to the bits-per-dimension of the learned circuit. Over 5 random restarts, the stochastic WM algorithm resulted in a Wasserstein objective between 29947 and 29986; conversely, the deterministic WM algorithm resulted in a Wasserstein objective of 32766. However, this decrease in Wasserstein distance resulted in no decrease in bits-per-dimension for the trained models, with stochastic WM yielding circuits with BPDs of between 1.503 and 1.537. See Table 1 for more details.

## D.3 VISUALIZING TRANSPORT PLANS BETWEEN PCS

Since our algorithm does not only return $CW_p$ between two circuits but also the corresponding transport plan, we can visualize the transport of point densities between the two distributions by conditioning the coupling circuit on an assignment of random variables in one circuit. We can similarly visualize the transport plan for an arbitrary region in one PC to another by conditioning on the random variable assignments being within said region.

Since the transport plan for a single point (or a region of points) is itself a PC, we can query it like we would any other circuit; for example, sampling a set of corresponding points, as well as computing *maximum a posteriori*—which is tractable if the original two circuits are marginal-deterministic (Choi et al., 2020)—for the transport plan of a point corresponds to the most likely corresponding point in the second distribution for the given point. Because a coupling circuit inherits the structural properties of the original circuit, it is straightforward to understand what queries are and are not tractable for a point transport map.

In Figure 8, we provide an example of visualizing the optimal transport plan between two randomly-generated PCs. We note that despite the transport plan being constrained to be a PC with a certain structure, the resulting transport plan matches our intuition as to what an optimal transport plan should look like.

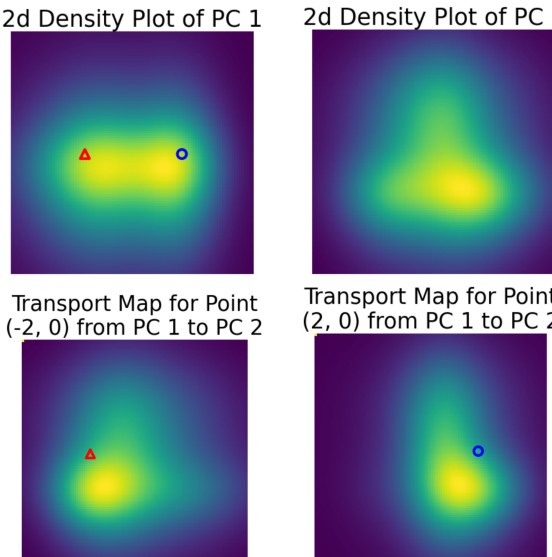

Figure 8: Visualization of transporting the indicated points from the distribution parameterized by PC 1 to the distribution parameterized by PC 2. The points (red triangle and blue circle) were arbitrarily selected to show how a point mass is redistributed according to the computed transport map. The top two figures visualize the input distributions, while the bottom two figures visualize where the point density indicated is transported to from the first to the second distribution.

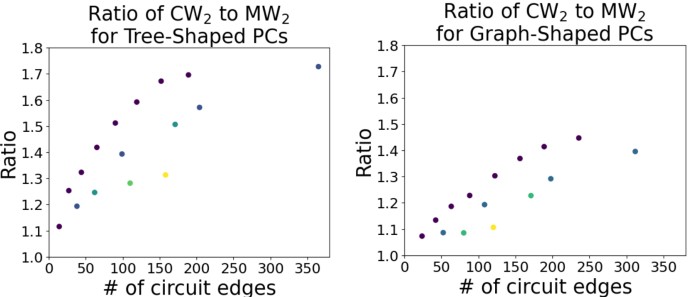

Figure 9: Ratio of $\frac{CW_2}{MW_2}$, lower is better. Each data point represents a pair of circuit structures corresponding to a fixed circuit branching factor and fixed number of random variables in the circuit scope, averaged over 100 random parameter initializations. Empirically, the gap between $CW_2$ and $MW_2$ grows roughly linearly in the size of the circuit. The hue of each point represents the circuit depth, with lighter points being a higher depth.

## D.4 EMPIRICALLY COMPARING THE QUANTITIES $CW_2$ AND $MW_2$

Figure 9 shows with larger circuits having a larger difference between $CW_2$ and $MW_2$ when compared to smaller circuits, corresponding to $CW_p$ being a looser upper-bound of both $MW_p$ and the true Wasserstein distance.

