# OpenReview forum: "Optimal Transport for Probabilistic Circuits"
_ICLR.cc/2025/Conference — Submitted to ICLR 2025_

### Official Review · Reviewer_FxdE · 2024-10-17

**Soundness:** 3
**Presentation:** 4
**Contribution:** 4
**Rating:** 8
**Confidence:** 3

**Summary:**

The authors consider the computational complexity of Wasserstein-type distance metrics between probabilistic circuits (PCs), and show interesting and novel algorithms and lower bounds (hardness results) for computing them.
Notably, they show that it is $\sf NP$-hard to compute the $\infty$-Wasserstein distance between PCs, and there is an efficient algorithm for computing the Circuit Wasserstein distance between PCs.

**Strengths:**

The choice of the problem and the strength of the results.

**Weaknesses:**

See the questions below.

**Questions:**

Page 1:
Please consider these papers *(and their many relevant references therein)* about the computational aspects of total variation distance:

[1] Bhattacharyya, A., Gayen, S., Meel, K. S., Myrisiotis, D., Pavan, A., and Vinodchandran, N. V.: On approximating total variation distance. In Proc. of IJCAI, pp. 3479–3487. ijcai.org, 2023. Links: https://arxiv.org/abs/2206.07209; https://www.ijcai.org/proceedings/2023/387.

[2] Weiming Feng, Heng Guo, Mark Jerrum, Jiaheng Wang: A simple polynomial-time approximation algorithm for the total variation distance between two product distributions. TheoretiCS 2 (2023). Link: https://arxiv.org/abs/2208.00740v3.

[3] Weiming Feng, Liqiang Liu, Tianren Liu: On Deterministically Approximating Total Variation Distance. SODA 2024: 1766-1791. Link: https://epubs.siam.org/doi/10.1137/1.9781611977912.70.

[4] Arnab Bhattacharyya, Sutanu Gayen, Kuldeep S. Meel, Dimitrios Myrisiotis, A. Pavan, N. V. Vinodchandran: Total Variation Distance Meets Probabilistic Inference. Link: https://arxiv.org/abs/2309.09134.

Page 2:
Can you please further elaborate on the notions of smoothness and decomposability?

Page 3:
Please elaborate on the caption of Figure 1.

Page 4:
Section 3.2:
You SHOULD put your algorithm in a theorem statement :)

Page 5:
Please define the methods you use in Algorithm 1.

Lines 239 -- 255:
This part should be more detailed :)

Page 6:
Section 4.1:
Please use a theorem statement for your algorithm.

I do not understand Equation (4):
Why is the second equality correct?

Page 7:
Can you please elaborate on Lines 347 -- 352?

Page 8:
I am not an expert with experiments :)

Page 10:
Can you please add more details to the future work part, etc.?
It looks too small now.

---

> ### Author Response · Authors · 2024-11-15
>
> Thank you for your feedback. We have revised the paper based on your review, including: rewriting Algorithm 1 and the following description to be more clear, clarifying the parallels between EM and our algorithm WM, and adding details about future work.
> Thank you for the pointers to related works. We have revised the introduction to discuss related work regarding the TV distance.
>
> > … Equation (4): Why is the second equality correct?
>
> The second equality in Equation 4 makes use of how the empirical distribution \hat{Q} is defined; we have updated the notation of this section and clarified this in the paper.
>
> Along with the above changes, we have also clarified some sections and modified some formatting according to your comments. Thank you again for your feedback.

---

> ### Author Response · Authors · 2024-11-25
>
> We hope that we have addressed your comments in the latest revision of our paper. As the revision period is coming to an end soon, please let us know if you have any unaddressed questions or suggestions for us to improve the paper.

---

> > ### Comment · Reviewer_FxdE · 2024-11-27
> >
> > Thank you very much! I will keep my positive score :)

---

### Official Review · Reviewer_d4Zo · 2024-10-28

**Soundness:** 3
**Presentation:** 3
**Contribution:** 3
**Rating:** 5
**Confidence:** 3

**Summary:**

This paper explores approaches for computing and bounding the Wasserstein distance and optimal transport plans in two settings: between two probabilistic circuits and between a probabilistic circuit and an empirical distribution. For the former, it introduces a Wasserstein-type distance that upper-bounds the true Wasserstein distance and provides an efficient algorithm for exact computation. For the latter, the authors present a parameter estimation algorithm designed to minimize the Wasserstein distance between a circuit and an empirical distribution. The proposed methods are validated through empirical evaluations on both randomly generated probabilistic circuits and a benchmark dataset.

**Strengths:**

This paper introduces, for the first time, a Circuit Wasserstein distance, denoted as $ CW_p $, between compatible probabilistic circuits (PCs). Leveraging the recursive properties of the Wasserstein objective, it computes the optimal parameters for the coupling circuit by solving a small linear program at each sum node. Additionally, the paper presents a method for learning the parameters of PCs by minimizing $ ECW_p $, which is computationally efficient.

**Weaknesses:**

This paper addresses only a restricted set of cases within the broader context of probabilistic circuits. Regarding optimal transport, the approach feels somewhat formulaic, lacking a deeper exploration of the essential relationship between optimal transport and probabilistic circuits. Additionally, there is a typo on line 786.

**Questions:**

1. Could you provide additional clarification on the proof of Theorem 1 and 2? I think it's better to add some graph in the proof.
2. Have you evaluated $CW_p$ across a broader variety of PCs, as its application might be limited?
3. Could you elaborate on the computational complexity of your approach?

---

> ### Author Response · Authors · 2024-11-15
>
> Thank you for your feedback on the paper; we have made some revisions, and will follow up with another revision addressing the remaining feedback shortly.
>
> > Q1. Could you provide additional clarification on the proof of Theorem 1 and 2? I think it's better to add some graph in the proof.
>
> We will include figures in the appendix to clarify the proofs of Theorems 1 and 2 shortly.
>
> > Q2. Have you evaluated CWp across a broader variety of PCs, as its application might be limited?
>
> We are working on experiments that utilize optimal transport maps between PCs for color transfer between images to showcase an interesting and practical application of our algorithm, as well as computing the optimal transport distance between larger circuits learned on high-dimensional image data.
>
> > Q3. Could you elaborate on the computational complexity of your approach?
>
> We compute CW2 and the associated transport map in $\mathcal{O}(mn)$-time, where $m$ and $n$ are the number of edges in each original circuit (see the last paragraph of Section 3.2 for more details). We show that computing the circuit parameters that minimize the empirical Wasserstein distance is NP-hard (see Theorem 2), but our proposed iterative algorithm where each step runs in $O(n)$-time in the number of circuit edges is guaranteed to converge to a local minimum (see the second-to-last paragraph of Section 4.1 for more details).
>
> > This paper addresses only a restricted set of cases within the broader context of probabilistic circuits.
>
> Within the framework of probabilistic circuits, the tractability of certain queries is guaranteed by imposing constraints on the circuit structure (smoothness, decomposability, and compatibility are required in our case). We also note that the constraints we impose for our algorithm are the same constraints required by all existing tractable algorithms for pairwise queries (such as KL-divergence (Vergari et al., 2021)) between PCs. Crucially, enforcing these structural properties do not restrict the PCs’ expressivity (i.e., they can still represent any distribution), but may limit their expressive efficiency (i.e., the circuit may need to be exponentially large). Our algorithm is thus applicable to any two PCs - although we incur a possibly exponential increase in the size of the circuits by making them compatible. We appreciate your comment, and have noted this in the paper.
>
> Additionally to the changes mentioned above, we have fixed the typo on line 786. Thank you again for your feedback.

---

> ### Author Response · Authors · 2024-11-25
>
> We hope that we have addressed your comments in the latest revision of our paper. As the revision period is coming to an end soon, please let us know if you have any unaddressed questions or suggestions for us to improve the paper.

---

### Official Review · Reviewer_mehN · 2024-11-01

**Soundness:** 3
**Presentation:** 3
**Contribution:** 3
**Rating:** 6
**Confidence:** 3

**Summary:**

This paper focuses on computing (or bounding) the Wasserstein distance and optimal transport plan between (i) two probabilistic circuits and (ii) a probabilistic circuit and an empirical distribution. For (i) a Wasserstein-type distance that upper-bounds the true Wasserstein distance was proposed and provided an efficient and exact algorithm for computing it between two circuits. For (ii) a parameter estimation algorithm for PCs that seeks to minimize the Wasserstein distance between a circuit and an empirical distribution was proposed.

**Strengths:**

1- The proofs of the theorems are correct, and the mathematical accuracy is high.

**Weaknesses:**

1- Using simple examples to illustrate definitions and results could make the paper easier to read and follow.

2- While the proposed metrics could significantly reduce runtime, they also lead to an increase in error. How much error is considered acceptable? There is no analytical approach or numerical result provided to show the impact of this error.

3- The proposed method performs well with a small set of variables; however, runtime challenges typically arise in large-scale systems with many variables.

4- There are insufficient numerical results to illustrate all aspects of the proposed distance. Applying the method to practical problems and providing comparisons with other works in terms of runtime and accuracy would strengthen the paper.

5- The application of this metric is not clearly explained in the paper. Additionally, given the limitations of the proposed CW and ECW metrics—such as susceptibility to error and effective performance only with a small set of variables—the metric has limited applicability in practical, real-world problems.

**Questions:**

1- Why wasn’t CW compared with W?

2- How much error is considered acceptable?

---

> ### Author Response · Authors · 2024-11-15
>
> Thank you for your feedback on the paper; we have updated the existing example of a coupling circuit and will be adding example figures for some of the proofs shortly.
>
> > Q1. Why wasn’t CW compared with W?
>
> To the best of our knowledge, there are no existing algorithms that can be run for even very small circuits in a reasonable amount of time that exactly compute W. Thus, such a comparison is unfortunately quite difficult.
>
> > Q2. How much error is considered acceptable?
>
> To verify the applicability of our approach to real-world problems, we are working on experiments that utilize optimal transport maps between PCs for color transfer between images to showcase a practical application of our algorithm.
>
> > Using simple examples to illustrate definitions and results could make the paper easier to read and follow.
>
> We have updated Figure 1 to more clearly explain a coupling circuit.
>
> > The proposed method performs well with a small set of variables; however, runtime challenges typically arise in large-scale systems with many variables.
>
> We first note that Appendix C.1 contains experimental results for computing the transport map between circuits that are two orders of magnitude larger than those mentioned in the body of the paper, which were limited in size to be able to compute MW2. We are also working on experiments that compute the optimal transport distance between two circuits learned on high-dimensional image data.
>
> > While the proposed metrics could significantly reduce runtime, they also lead to an increase in error. How much error is considered acceptable? There is no analytical approach or numerical result provided to show the impact of this error.
>
> In Section 5.2, we perform experiments to quantify the error between CW2 and MW2 as the ratio between the two quantities. Unfortunately, we are unable to include more data points in these figures, as computing MW2 becomes impractical for circuits larger than those plotted.
> We are open to suggestions on additional numerical results that we could provide to showcase the gap between MW2 and CW2.

---

> > ### Comment · Reviewer_mehN · 2024-11-25
> >
> > Thank you for revising the paper and addressing the questions. The paper has improved, and I found responses to several of my concerns. I have decided to update my score to 6. Of course, I am still concerned about the trade-off between runtime and accuracy, because there is no theoretical bound for the estimation error.

---

### Official Review · Reviewer_WAoc · 2024-11-04

**Soundness:** 2
**Presentation:** 4
**Contribution:** 2
**Rating:** 3
**Confidence:** 4

**Summary:**

The authors define (and show that it is tractable to compute) an analogue of Wasserstein distance (CW) between distributions encoded by structurally-identical probabilistic circuits (PCs). The high-level idea is to restrict to transport maps that have an analogous structure, and recursively decompose them, through which one obtains a metric that is an upper bound on the Wasserstein distance between the underlying measures.  The authors also provide a tractable analogue of a Wasserstein distance between a PC and an empirical distribution.  Code is provide for both procedures.  Random PCs are then generated according to a fixed structure, and the gap between CW and MW (another quantity in the literature) is studied, along with runtime. They also evaluate their measure of distance between PC and an emperical distribution against the EM algorithm in a learning context.

**Strengths:**

The ideas in this paper come through clearly, as the text (if not the math) is well-written.  The general idea is natural, and it is not hard to see how a more efficient way of calculating Wasserstein distances between the distributions encoded by PCs could, in principle, be tremendously useful.  Propositions 1 and 2 provide important grounding for the given constructions.  The authors take the space needed to unpack their ideas.  They identify appropriate baselines, and some genuine (if perhaps insufficient) effort has been made to empirically evaluate these ideas.  The proofs are present and look ok at a very high level (although I only skimmed parts of them).

**Weaknesses:**

Unfortunately, the paper is lacking in technical depth.  The idea and its implementation seem straightforward to me, and the results are unsurprising --- so I find the mathematical contribution relatively small (putting aside the substantial low-level issues with it, which I detail below).  This could easily be forgiven if the techniques enable something really interesting, but, on the experimental side, the examples are all small-scale synthetic toys. To the extent that this is really a basis of a sold paper, I believe it requires either an interesting application, a more interesting theorem, or to show that there are conceptual/instrumental benefits to using this approach. Finally, I also believe that the limitations of this approach are not well-enough explored (or indeed, well-enough advertised at the beginning of the paper).  Specifically, the fact that this distance measure only applies to PCs with exactly the same structure, is an enormous shortcoming which does not come through at the beginning.  I therefore believe that the technical contribution is being significantly oversold.

The empirical results are not particularly encouraging.  The most obvious application of this paper would be to use this novel, easily-computed Wasserstein analogue for fitting parameters for PCs.  The authors correctly identify that their metric is closely related to EM, and compare against it as  baseline. However, their method does significantly worse. Yet the reasons for this are not discussed, and the authors do not mention any compensatory strengths that their distance measure has.  After finishing section 5.4, a reader can't help but wonder: why not just use EM for this task?


As for the presentation, my biggest complaint is that the math is not handled very deftly.  While presentation and the mathematical English looks great a glance, there are a number of places where it is ambiguous or unclear or technically wrong.  I have detailed a number of significant zones below that look like train wrecks to me (but they are all salvageable, if one were to invest some significant effort).


--- detailed comments below ----

Definition 1 is sloppy; it is mathematically imprecise and ambiguous.  What is the relationship between **X** and the nodes of the DAG?  How are the parameters for the sum nodes normalized?  What does the univariate probability distribution have to do with the variables?  Is the root of the dag a source or a sink, or possibly neither? The notion of scope, which is clearly important for the definitions to follow, is not defined at all.  Thus, when we get to the equation on line 097, I am very confused.  I do not know for certain which node is supposed to define a distribution over **X** (although I can only assume it is the root).  It also makes it seem that "input nodes" must correspond to variables of X.  But I'm still not clear on whether this needs to be a bijection or not.  Finally, additional restrictions are required to ensure that the result is actually a probability density function.  At the least, for product nodes, children must have disjoint supports (a property which, after definition 1, the authors call "decomposability"). But decomposability is not just required for the results of the paper results; it is required even for the words "define a probability distribution" to be correct. Either way, the current formalism doesn't typecheck; it implicitly eliding a projection in the definition of p_n for product and sum nodes.  Had I not seen this definition before, I would have been incredibly lost.

(Line 107) Footnote 1: there's no need to "abuse notation". Simply choose the appropriate base measure, and use the radon-nikodym derivative.

Algorithm 1 has some problems.
 - the procedure cache (n, m) is not defined.
 - there is no need to build the LP constraints iteratively with logic in the algorithm; just define the problem mathematically, and say "solve problem (P)"
 - Line 228 makes no sense to me.  What are the semantics of running LP.objective <- (...) multiple times?
 - the sum over _{i,j}  seems to make the iteration over \theta_{i,j} in r.params useless.
 - the fields "-.params" are not defined.
To summarize: this is not an algorithm---it is a snippet of Python code removed from its context and made slightly more colloquial. To call it an "algorithm", everything has to refer to something defined mathematically in the text!

Equation (4) exposes a deep flaw in the chosen notation: in the subscript of the expectation, there is not distinction between {\bf x}, which is bound by the expectation operator, and k, which is bound by the sum earlier.  In fact, the distribution you're taking an expectation over is should really be \gamma( x | k), not \gamma(x,k).  The same comment holds for the formula in definition 5.

Line 282. The fact that y^k ~ Q and the i.i.d assumption are irrelevant for the definition.  You can just start with arbitrary y^k and define the empirical distribution \hat Q from it.

**Questions:**

1.  The notion of compatibility seems really quite strong. Are there any ways you can see of weakening it, so that structures that are similar (but not quite identical) can still be handled with your methods? It would be particularly nice if the method were to degrade naturally to an intractable (exponentially-sized) problem in the worst case when the two PCs have very different structures.  Do the authors think this might be possible?

2.  Theorem 1 says that computing the \infty-Wasserstein distance is coNP-hard.  But this explicitly leaves open the possibility that calculating W1 or W2 is easier.  W1 and W2 are the most commonly-used in practice, and the motivation for this work has been that calculating Wp is hard.  What is the obstacle to showing that calculating W1 or W2 is hard?  If it is interesting, it is worth highlighting.  However, if the authors think that it might be possible to find a tractable algorithm for W1 or W2, then I feel strongly that it would be better for the research community to delay publication until this question is resolved.

3.  Proposition 2 states that W(P,Q) <= CW(P,Q).  In light of the comments on the top of page 6 and the bottom of page 8, it seems that, in addition, W(P,Q) <= MW(P,Q) <= CW(P,Q).  Thus, it may be worth providing a framing this paper (at least locally, where this is discussed) as a looser and faster approximation MW.  In what contexts are MW used? Has it been experimentally validated? Might you be able to show a computational benefit in prior tasks by drawing from that literature?

4. I found the Figure 4, and its explanation on lines 459-462 difficult to understand. I believe there is a missing word ("of"?) on line 462, but I can't figure out what the significance of this visualization is. What is the point of displaying this particular transport plan? What does this have to do with the story of the paper?

5.  I assume that the points in Figure 3 represent pairs randomly sampled circuits. Are they all with the same architecture, or do only pairs have the same architecture?  Why are there so few points?    Figure 3 suggests to me that your method may work well on circuits of larger depths.   Have you tried to quantify this across a broader set of distributions over {P,Q}, or theoretically show that this must be the case?  These might be interesting avenues to explore.

---

> ### Author Response · Authors · 2024-11-15
>
> Thank you for your valuable feedback on the paper; we appreciate the time and effort you put into reviewing our work.
>
> > Q1. The notion of compatibility seems really quite strong. Are there any ways you can see of weakening it…? It would be particularly nice if the method were to degrade naturally to an intractable (exponentially-sized) problem in the worst case when the two PCs have very different structures…
>
> We would like to first clarify that compatibility does not require that two circuits have identical structure; it only requires that two corresponding product nodes with same scopes decompose the scopes in the same way into children (up to a bijection in our case). In other words, two compatible circuits have the same hierarchical scope partitioning, but they can have different structures. Furthermore, compatibility between two circuits is necessary for all pairwise queries (such as KL-divergence or joint entropy (Vergari et al., 2021)) with known tractable algorithms for PCs as far as we can tell. Crucially, enforcing these structural properties do not restrict the PCs’ expressivity (i.e., they can still represent any distribution), but may limit their expressive efficiency (i.e., the circuit may need to be exponentially large). Thus, our algorithm can already be applied to arbitrary pairs of PCs with very different structures, although we incur a possibly exponential increase in the size of the circuits by making them compatible. We appreciate your comment, and have clarified this in Section 3.1 and Figure 1 in the paper.
>
> > Q2. …W1 and W2 are the most commonly-used in practice … if the authors think that it might be possible to find a tractable algorithm for W1 or W2, then I feel strongly that it would be better for the research community to delay publication until this question is resolved.
>
> While W1 and W2 are the most common in practice, they are generally approximated. Moreover, even though there are various approximation methods for W2 between GMMs, the question of whether exactly computing W1 or W2 between GMMs (which are a special case of PCs) is NP-hard is still an open question to the best of our knowledge. Thus, we consider this result outside the scope of this work–introducing the first algorithm to compute and optimize Wasserstein-type distances for PCs–and respectfully disagree that publication of this work should be delayed until we can resolve this question.
>
> > Q3. … In what contexts are MW used? Has it been experimentally validated? Might you be able to show a computational benefit in prior tasks by drawing from that literature?
>
> Past work has experimentally validated MW and associated transport maps when applied to the task of color transfer (Delon & Desolneux, 2020). We are currently working on applying our optimal transport approach to the task of color transfer, which will provide a direct comparison between CW and MW for a real-world application. This will complement our existing results showing the computational benefit of CW over MW between synthetic PCs. We hope to follow up soon with the results.
>
> > Q4. …significance of [the Figure 4] visualization…
>
> We include figure 4 to a) demonstrate that we can easily get the transport plan from our CW algorithm and b) show that the transport plan matches our intuition for what the optimal transport plan should be, despite it being an approximation. However, in the interest of space in the body of the paper, we have moved the figure to the appendix.
>
> > Q5. I assume that the points in Figure 3 represent pairs randomly sampled circuits. Are they all with the same architecture, or do only pairs have the same architecture? Why are there so few points? Figure 3 suggests to me that your method may work well on circuits of larger depths. Have you tried to quantify this across a broader set of distributions over {P,Q}, or theoretically show that this must be the case? These might be interesting avenues to explore.”
>
> We apologize for the confusion regarding our experiment setup, and have clarified in the paper that each point represents an average over 100 randomly-initialized circuits with a fixed dimensionality and sum node branching factor. We have also provided justification on why our approach seems to work well for circuits with a higher depth; in short, since we only plot the number of circuit edges rather than the number of learnable parameters in the circuit (which are different since product node edges are unweighted), it is possible for a higher-depth circuit to have more edges but less learnable parameters than a lower-depth circuit and thus incur less error. Lastly, we are unable to include more data points in these figures, as computing MW2 becomes impractical for circuits larger than those plotted.
>
> Due to the OpenReview character limit, we answer the remaining comments in our second reply.

---

> > ### Comment · Reviewer_WAoc · 2024-11-21
> >
> > I'd like to thank the authors for putting in significant effort to answer my questions and address my comments.
> >
> > ## Q1.
> >
> > I am increasingly confused about circuit compatibility.
> >
> > > We would like to first clarify that compatibility does not require that two circuits have identical structure; it only requires that two corresponding product nodes with same scopes decompose the scopes in the same way into children (up to a bijection in our case). In other words, two compatible circuits have the same hierarchical scope partitioning, but they can have different structures.
> >
> > I think that is what I meant. I believe there is an equivalence of structures modulo what you call hierarchical scope partitioning, and it seems like having the same equivalence class in that sense is a very strong assumption.
> >
> > > Crucially, enforcing these structural properties do not restrict the PCs’ expressivity (i.e., they can still represent any distribution), but may limit their expressive efficiency (i.e., the circuit may need to be exponentially large). Thus, our algorithm can already be applied to arbitrary pairs of PCs with very different structures, although we incur a possibly exponential increase in the size of the circuits by making them compatible.
> >
> > Sorry, I'm afraid I didn't follow the reasoning.
> > How, exactly, does one show that an arbitrary pair of PCs can be rewritten in a way such that they are decomposable (possibly incuring exponential circuit size)? The Choi 2020 reference does not mention compatibility (so that reference is misplaced in the new material). The Vergari paper Figure 2 has helped suggest to me that this might be possible, but I could not find any place where they show this. The gloss does mention "polynomial time" in their informal gloss of compatibility (page 5), but their definition (Defn 2.5) does not.
> >
> > Answering this related question might help my understanding: suppose $\bf X$ is a set of $n$ binary variables, and $P({\bf X})$ is an arbitrary tensor of shape $(2,2,2, \ldots,2)$ with no additional structure such as (in)dependencies.  Can it be represented by an (exponentially large) "universal" circuit? If so, is that circuit structurally compatible with all other circuits?
> >
> >
> > ## Q2.
> >
> > Let me try to explain my complaint in a different way: I think Theorem 1 feels sneaky, and there is a sense in which its current presentation weakens the paper.
> > The motivation is for Wasserstein distances in general, but naturally focus more on W1 and W2.  The experiments focus exclusively on W2.
> > Yet the hardness results focus on a very particular special case, which intuitively might be especially hard.
> > I think some deference to the gap here (and explicit acknowledgement that the hardness for other $p$ is an open question) is necessary.
> > Of course, if you could resolve that open problem, it this would be  very strong motivation for your approach in the next sections.
> > Currently the connection between Theorem 1 and the rest of the paper seems a bit tenuous, and the conceptual gap has been swept under the rug.
> >
> > ## Comparison with EM
> >
> > I still don't buy the significance of the experiments.  The distinction between the numbers in Figure 6 (which is technically a table) do not appear to support the claim that the proposed method "outperforms EM with regard to learning circuits with a lower Wasserstein distance to the empirical data distribution", at least not with any kind of significance.
> >
> > Also, as far as proof-of-concept for application, likelihood is more important than W2 to the empirical data distribution.  I maintain that the experimental evaluation is weak.
> >
> >
> > ## Interim Summary
> >
> >
> > In general, thank you for the updates! I think they have improved the paper. I am not certain that they go far enough, but I will look again at the final version and reconsider my score after the discussion period.
> >
> > To summarize my thoughts at the moment: I think this idea has potential, but it has not yet been fully actualized. The pieces are there, but it remains to show a deep theoretical result, or to properly establish a proof-of-concept in which this approach offers a practical advantage over others. I think publishing this paper too early could blunt its impact.

---

> > > ### Author Response · Authors · 2024-11-25
> > >
> > > Thank you for your response. We have uploaded a revised version of the paper addressing the feedback below.
> > >
> > > ## Q1
> > >
> > > > How, exactly, does one show that an arbitrary pair of PCs can be rewritten in a way such that they are decomposable (possibly incurring exponential circuit size)? The Choi 2020 reference does not mention compatibility (so that reference is misplaced in the new material).
> > >
> > > We have elaborated on making two arbitrary circuits compatible in the paper and included new references to support this. Succinctly, an arbitrary circuit can be made structured-decomposable while incurring a possibly exponential increase in circuit size (de Colnet & Mengel 2021), and two incompatible structured-decomposable circuits can be made compatible while incurring yet another potentially exponential increase in circuit size (Zhang et al. 2024).
> > > Despite this in-practice large increase in circuit size when making two circuits compatible, we would like to note that competitive structure learning algorithms such as Strudel (Dang et al. 2020) can be utilized to learn a structure respecting a specific vtree (hierarchical scope partitioning), allowing one to direct learn compatible circuits as we do in our experiments.
> > >
> > > > Answering this related question might help my understanding: suppose $\mathbf{X}$ is a set of binary variables, and $P(\mathbf{X})$ is an arbitrary tensor of shape $(2,2,2,...,2)$ with no additional structure such as (in)dependencies. Can it be represented by an (exponentially large) "universal" circuit? If so, is that circuit [$P(\mathbf{X})$] structurally compatible with all other circuits?
> > >
> > > Yes, it can be represented by a PC that is a root sum node with $2^n$ product nodes for children, with each product node having $n$ univariate input node children. Each child of the sum node corresponds to one of the $2^n$ variable assignments; the edge weight corresponding to this child is $P(\mathbf{X})$. Such circuit is called omni-compatible, as its product nodes can easily be rearranged to make it compatible with any decomposable circuit over the same scope.
> > >
> > > ## Q2
> > >
> > > We agree that our presentation of Theorem 1 could have been misleading, and have revised the paper to clarify its purpose. Our goal is to create an algorithm that can solve for or upper-bound the $p$-Wasserstein distance for arbitrary $p$; while there may still be an efficient algorithm for some other $p$, Theorem 1 showing that computing $W_\infty$ exactly between circuits is coNP-hard. Therefore, the motivation for proposing $CW_p$ comes from its tractability for all $p$, which cannot be said for $W_p$. We have clarified this in the paper.
> > >
> > > ## Comparison with EM
> > >
> > > We will update the paper with stronger experimental evaluations of Wasserstein learning shortly.
> > >
> > > Thank you again for your valuable feedback.

---

> ### Author Response · Authors · 2024-11-15
>
> We answer the remaining comments below.
>
> > on the experimental side, the examples are all small-scale synthetic toys.
>
> We first note that experimental comparisons for distance computation in the main paper (Figs 2 & 3) were done on smaller circuits because the baseline (MW) could not scale to larger circuits. Appendix C.1 contains experimental results for computing the transport map between circuits that are two orders of magnitude larger, albeit still synthetic. Moreover, the parameter learning experiments trained circuits with up to millions of parameters using MNIST benchmark dataset. Nevertheless, we agree that additional experiments with real-world applications would be valuable. We are currently exploring utilizing optimal transport maps between PCs for color transfer between images, as well as computing the optimal transport distance between two circuits learned on high-dimensional image data.
>
> > The authors correctly identify that their metric is closely related to EM, and compare against it as baseline. However, their method does significantly worse.
>
> While our approach achieves worse likelihoods than EM (which explicitly maximizes likelihood), we note that our approach outperforms EM with regard to learning circuits with a lower Wasserstein distance to the empirical data distribution (experimental results in Appendix C.2). We are currently investigating a slight variation of our current algorithm that appears to yield circuits with similar likelihoods to our current approach but significantly lower Wasserstein distances to the empirical data distribution, which will be included in the paper shortly.
>
>
> Lastly, thank you for your detailed comments about definitions and notations. We mostly followed the standard definitions for PCs (Choi et al., 2020) but agree that they could be made more precise and have revised the paper accordingly.

---

### Author Response · Authors · 2024-11-25

As promised, we have revised the experiments section of the paper to include color transfer experiments and more robust $CW_2$ computation experiments between learned (non-synthetic) circuits. Changes since our last revision are highlighted in green.

---

### Author Response · Authors · 2024-11-27

We have made our final changes to the paper as the revision period comes to a close, including additional experimental data for Wasserstein learning in Appendix D.2. While the changes are no longer highlighted, reviewers can view the revision history of the paper to bring up a version where the changes were highlighted. We appreciate all of the feedback provided by the reviewers, and welcome additional discussion during the remainder of the discussion period.

---

### Meta-Review · Area_Chair_WG6m · 2024-12-20

**Metareview:**

The paper introduces a tractable analogue of Wasserstein distance (CW) for structurally-identical probabilistic circuits and provides an algorithm for its computation, but empirical evaluation shows limited practical advantages. The paper is well-written and presents a novel approach to computing this distance for PCs, with clear theoretical grounding and efficient algorithms. The proposed CW distance has limited applicability due to the strong requirement of structural compatibility between PCs. The reviewers overall did not find empirical results to be convincing, particularly in comparison to simpler existing methods.

The limited applicability of the proposed method, combined with unconvincing empirical results and lack of clear practical advantages over existing techniques, were the main points raised by reviewers. Compared to other submission, there was no strong enthusiasm in favor of this work.

**Additional Comments On Reviewer Discussion:**

The reviewers all communicated with the authors during the discussion phase, but were not ultimately convinced.

---

### Decision · Program_Chairs · 2025-01-22

Reject